# Innovation in Additive Manufacturing Using Polymers: A Survey on the Technological and Material Developments

**DOI:** 10.3390/polym14071351

**Published:** 2022-03-26

**Authors:** Mauricio A. Sarabia-Vallejos, Fernando E. Rodríguez-Umanzor, Carmen M. González-Henríquez, Juan Rodríguez-Hernández

**Affiliations:** 1Facultad de Ingeniería y Tecnología, Universidad San Sebastián, Sede Santiago, Santiago 8420524, Chile; mauricio.sarabia@uss.cl; 2Facultad de Ciencias Naturales, Matemáticas y del Medio Ambiente, Departamento de Química, Universidad Tecnológica Metropolitana, Santiago 7800003, Chile; guezu@utem.cl; 3Programa Doctorado en Ciencia de Materiales e Ingeniería de Procesos, Universidad Tecnológica Metropolitana, Santiago 8940000, Chile; 4Programa Institucional de Fomento a la Investigación, Desarrollo e Innovación, Universidad Tecnológica Metropolitana, Santiago 8940000, Chile; 5Polymer Functionalization Group, Departamento de Química Macromolecular Aplicada, Instituto de Ciencia y Tecnología de Polímeros-Consejo Superior de Investigaciones Científicas (ICTP-CSIC), 28006 Madrid, Spain; jrodriguez@ictp.csic.es

**Keywords:** additive manufacturing, material extrusion, fused deposition modeling, selective laser sintering, stereolithography, multimaterial 3D printing

## Abstract

This review summarizes the most recent advances from technological and physico-chemical perspectives to improve several remaining issues in polymeric materials’ additive manufacturing (AM). Without a doubt, AM is experimenting with significant progress due to technological innovations that are currently advancing. In this context, the state-of-the-art considers both research areas as working separately and contributing to developing the different AM technologies. First, AM techniques’ advantages and current limitations are analyzed and discussed. A detailed overview of the efforts made to improve the two most extensively employed techniques, i.e., material extrusion and VAT-photopolymerization, is presented. Aspects such as the part size, the possibility of producing parts in a continuous process, the improvement of the fabrication time, the reduction of the use of supports, and the fabrication of components using more than one material are analyzed. The last part of this review complements these technological advances with a general overview of the innovations made from a material perspective. The use of reinforced polymers, the preparation of adapted high-temperature materials, or even the fabrication of metallic and ceramic parts using polymers as supports are considered. Finally, the use of smart materials that enable the fabrication of shape-changing 3D objects and sustainable materials will also be explored.

## 1. Introduction

Additive manufacturing (AM) has been defined as joining materials layer-by-layer to make 3D parts [1]. The first method to create a three-dimensional object using CAD was rapid prototyping, developed in the 1980s to produce models and prototype parts. AM, also known as 3D printing (3DP), has significant advantages over other manufacturing processes such as milling or molding. For instance, AM significantly reduces tooling and has an astonishing ability to create almost any possible geometry.

Moreover, it is worth mentioning that the expiration of the original patents and the advances in different AM technologies have made it possible to acquire home-use 3D printers at accessible prices [2]. As a result, AM permits today both private and industrial users to design and produce their goods [2], supporting Toffler’s idea [3] of the prosumer rise (the same subject is a producer and consumer of the product), which increases the competitive threat proposed by AM technologies to the established firms [4]. Figure 1 depicts the areas where 3D printing has been applied so far, with their corresponding percentage respecting the whole market, subdivided into three main sectors, pre-production, production, and post-production. As can be observed, rapid prototyping, with nearly 25% of the total, is by far the most relevant application of 3D printing today. However, the percentage devoted to product development and direct manufacturing gradually increases, indicating a transition from limited 3D printing only for prototyping purposes for a wide range of applications. 

## 2. General Overview of the AM Methodologies Using Polymers: Current Advantages and Limitations

Today, a vast myriad of AM technologies has been developed depending on the type and form of material employed (powder, liquid/gel, or filament), the deposition system, or the source of energy used (heat, laser, or UV-light). ASTM International, American Society for Testing and Materials, implemented criteria for classifying AM technologies. Currently, the different technologies can be classified into seven different categories: (a) material extrusion, (b) powder bed fusion, (c) vat photopolymerization, (d) material jetting, (e) binder jetting, (f) sheet lamination, and (g) directed energy deposition [6,7]. 

It is worthwhile to mention that, while it is true that all these technologies are currently commercially available, the number of manufacturers and the market of each equipment significantly varies depending on the technology. As illustrated in Figure 2, machine sales are expected to increase by around 13% annually. More interestingly, this report has observed that material extrusion and, in particular, FFF (fused filament fabrication, or fused deposition modeling, FDM) is the most extensively employed technology. Vat photopolymerization, including SLA (stereolithography), DLP (digital light processing), and CDLP (continuous DLP), is predicted to grow in the following years and even overpass the FFF market. 

Nowadays, AM offers significant advantages over other currently employed manufacturing technologies [8] in some particular aspects. This method permits the fabrication of fully customized products with complex geometrical structures (internal or external) in an economical manner (primarily for limited productions) [9]. By using this technology, patterns can be easily created, personalized, and modified according to any requirement provided by the final user. This methodology also allows sharing the design so that the manufacturing process can be easily carried out in many different places simultaneously. 

Finally, it is worth mentioning that AM offers significant manufacturing process developments, particularly its environmental implications. This technique is more efficient in feeding materials and permits an environment-friendly design. Typically, lower energy is required for AM in comparison to subtractive processes [10]. In addition, it does not need chemicals during the manufacturing and allows, in most cases, the reuse of the feeding material—however, current significant limitations still restrict their use. 

Table 1 shows some characteristics of the three most extensively employed AM technologies. Some aspects that require improvement are found in the two technologies analyzed in this review, including the nature of the fabrication process, which is discontinuous, the part size, which is limited by the printer size, or the fabrication process, involving a layer-by-layer deposition. In addition to these, other aspects are specific to each technology. For instance, SLA and DLP do not currently allow the use of more than one material (in contrast to Polyjet^®^) and therefore, still require support for the fabrication of intricate structures. Material extrusion presents clear limits in terms of resolution and present anisotropic properties. 

Observing Table 1 is possible to conclude that there are several aspects in which FFF presents a clear superiority against SLA, and in other cases, it is the opposite. For example, regarding print quality and precision, which seems reflected in the resolution of each technology, SLA has the advantage, resolution as low as 20 μm in the Z-axis while for FFF is usually close to 200 μm (ten times better resolution). This happens because, in the case of FFF, the layers are deposited from molten polymeric material; therefore, the resolution of the part is mainly defined by the extrusion nozzle size. It is also expected that some layers may not fully adhere to one another, producing cues visible on the surface. On the other hand, SLA uses a liquid resin cured by a highly precise laser to form each layer, achieving more refined details. Another important aspect is the mechanical resistance of the material, which is mainly defined by two variables, firstly, the type of material used to print (which was posteriorly revised) and, secondly, the isotropic degree achieved by the printed method. SLA has a clear advantage over FFF in terms of isotropy, mainly because FFF printers produce a mechanical bond between layers. In contrast, SLA 3D printers create chemical bonds by cross-linking the photopolymers, resulting in fully dense parts. These bonds provide high degrees of lateral strength, meaning that the strength of the parts does not change with orientation. This makes SLA ideal for engineering and manufacturing applications where material properties matter.

But one of the main disadvantages that present SLA against FFF is the range of materials that could be used to fabricate parts. FFF printers work with a wide range of standard thermoplastic filaments, such as acrylonitrile butadiene styrene (ABS), polylactic acid (PLA), and various blends. In addition, engineering materials, such as polyethylene terephthalate glycol (PETG), polyamide (PA), or thermoplastic polyurethane (TPU), and high-performance thermoplastics such as polyether ether ketone (PEEK) or polyethyleneimine (PEI) are also available. On the other hand, although SLA has also considerably increased the range of materials used to print (clear resins, elastic resins, high-temperature resistance resins, and ceramic resins, among others), it is not yet at the level of development that FFF technology has. However, there has been an incredible advance in the fabrication and synthesis of new types of SLA resins in recent years. For example, the group of Barkane et al. [11,12] developed SLA resins derived from vegetable oils. They studied, through FTIR and photorheology measurements, the UV-curing of epoxidized acrylate from soybean oil (AESO)-based formulations. By adding appropriate functional comonomers, such as trimethylolpropane triacrylate (TMPTA) and 1,6-hexanediol diacrylate and adjusting the concentration of photoinitiator from 1% to 7%, they decreased the needed UV-irradiation time by up to 25%. DSC studies also show that the addition of appropriate reactive comonomers can increase glass transition temperature by 10 °C and thermal degradation temperature by 28 °C. Similar studies of SLA bio-based resins were developed in the last few years [13,14], thus demonstrating that SLA resin fabrication is still a field under development, and it could be a research niche for several applications.

Another critical aspect that needs to be considered in this comparison is the cost of the technology. First, the printers themselves have a different range of prices: while low-cost FFF printers could be found for less than USD 150, professional ones can be acquired for USD 2000 to USD 8000. On the other hand, professional SLA printers are more expensive, with values ranging between USD 3000 and USD 10,000. In terms of material costs, FFF also has an advantage with common values ranging around 50 USD/kg, while more specialized materials are between 100–150 USD/kg. In the case of SLA, the prices of the resins are higher, values ranging between 150–250 USD/lt. SLA can create parts up to 5 to 10 times faster than FDM 3D printers (using the same layer height, 200 μm) in printing speed. However, in terms of build volume, FDM could create much bigger parts using printers with similar costs. In Table 2, it is possible to observe a summary of some of these advantages and disadvantages commented on before.

As mentioned before, the type of material used in each printing technology (FFF or SLA) is fundamental for some applications where the chemical and mechanical response of the material matters. To be usable for FFF, the printing material must flow after fusion and then solidify. Thermoplastic polymers (preferably amorphous, crystalline ones that do not flow properly) are ideal for this application due to their low thermal expansion coefficient, glass transition temperature, and melting temperature. These properties can reduce internal stresses caused during cooling (such as warping, for example) [15]. Nowadays, a wide range of polymers is commercially available for FFF printing, the most common PLA and ABS. Nevertheless, other polymers have been equally employed in FFF processes, such as polycarbonate (PC), including PC-ABS blend and medical-grade PC, polycaprolactone (PCL), PA/Nylon, polyphenylsulfone (PPSF), or high-density polyethylene (HDPE) [16]. Additionally, high impact polystyrene (HIPS), TPU elastomer, and polyethylene terephthalate (PET) have also been used to fabricate parts with this technique for some novel applications. 

In the case of SLA, the selection of materials is more complex because the resin must combine several characteristics. At least three components are necessary to fabricate photosensitive resins for SLA: a photoinitiator, a multifunctional monomer/oligomer, and a diluent that can adjust the mixture’s viscosity. In general, the monomers/oligomers must have a relatively low-medium viscosity [17] to be able to form crosslinked polymers rapidly. The most common polymers used in SLA to fabricate 3D-printed parts are acrylates, in the case of free radical photopolymerization, and epoxy monomers, for cationic photopolymerization. 

In this context, this review aims to summarize the most recent advances from the technological and material point of view to tackle most of the limitations mentioned above. The technological advances will be discussed in the following sections related to the most extensively employed AM techniques, i.e., material extrusion and VAT photopolymerization. The technological advances selected for this bibliographic review mainly were focused on methodologies that imply a variety of existing technologies which can improve some relevant aspects of the technique, thus enhancing its performance. The selection was made in this way because we believe that these types of technologies are the most easily implementable today and are the ones that would generate the most significant impact on AM advancement. Section 5 is devoted to developments from the materials’ side. Thus, with selected examples, this section will illustrate the advances made to produce novel materials with additional properties adapted to AM to enlarge the range of applications of the final 3D-printed parts. 

## 3. Material Extrusion

Material extrusion (ME) is, without any doubt, the most employed AM technique for the fabrication of 3D-printed parts, either for domestic or industrial use. ME [18] includes those techniques in which a material is extruded through a nozzle by applying heat to melt a polymeric material, e.g., FFF, or pressure using viscoplastic pastes, e.g., direct ink writing (DIW) or bioprinting [19], among others. 

FFF is based on a thermoplastic filament heated over its Tg (glass transition temperature) and selectively extruded through a nozzle over a movable platform (build stage) to form a 3D object layer-by-layer. The main difference between FFF and DIW/bioprinting is that while the polymeric filaments are melted during extrusion (warmed over the glass transition temperature), the DIW/bioprinting systems rely purely on pastes with particular rheological properties stored in a syringe cartridge (Figure 3). 

While these two technologies are currently being extensively employed, several limiting aspects of these two AM methodologies are still under investigation. These restrictions include the size of the printer itself, the discontinuous fabrication that requires a manual operation after each printing, or the limitations in the extrusion system, which are directly related to the lower resolution compared to SLA. Herein, we will highlight some of the progress made to address these issues.

### 3.1. Overcoming the Size Limitations: Printing Parts Bigger than the Printer Itself

In general, the manufactured parts’ size is naturally bound to the size of their production machines. Interestingly, for 3DP, the ratio of machine volume vs. the produced part volume is closer to one in some cases, meaning that the 3D-printed parts could have almost the same volume as the machine in which they were printed. However, to date, most of the AM technologies do not allow fabrication of pieces larger than the printer itself. The development of manufacturing techniques that could increase this ratio (>1) should open many possibilities in the construction/architecture industry [21] or large-scale product fabrication [22].

Velez et al. [23] developed an innovative 3DP system that combines a climbing robot and an FFF printer. Coined as Koala3D printer, this machine could fabricate structures larger than its size by navigating vertically along the object being built. Its operation is based on two critical components: the printing head, which manufactures the printing objects (surrounded by a squared beam), and the climbing part, conformed by a robotic pair of synchronized actuated clamps attached to the squared beam. The idea is to convert this limited range of motion into an infinite range by changing the anchoring points to the beam. Figure 4 shows a diagram of the climbing mechanism and a photograph of the Koala 3D printer device. 

This type of technology could produce much wider objects that can overcome volumetric restrictions imposed by standard 3D printers, usable in application areas that require complex part fabrication with a high aspect ratio, such as aerospace or construction based on columns with intricate and complex inner structures. 

The development of structure-reconfiguring robots and collaborative constructions is one of the most widely studied concepts by several scientists related to manufacturing large architectural structures through 3D printing. For example, the MIT Mediated Matter Group (Media Lab-Massachusetts Institute of Technology, Massachusetts, USA) has created Fiberbots, an autonomous digital fabrication platform based on reconfigurable tubes designed to build large systems during a disaster quickly. The robots are mobile and use sensor feedback to control each tube’s length and curvature [24,25]. Similarly, Kovač et al. combined additive manufacturing techniques with aerial robotics. This research focused on developing a flying drone, or drone crews, capable of depositing polyurethane expanding foam in mid-flight to fabricate several types of structures [26,27]. Another interesting case was developed by Werfel et al., whose research was inspired by mound-building termites. It provides an example of an engineered complex system—many independent components—with multiple autonomous robots following an identical set of simple, local rules that collectively produce a specific structure requested by a user [28].

### 3.2. Non-Stop 3D Printing: Continuous Additive Manufacturing

In addition to the part size, a current major limitation is the discontinuity of the process, i.e., the part is fabricated and needs to be removed from the fabrication platform before the new printing process is started. The continuous 3DP or Z-infinite 3DP, based on a typical FFF apparatus, has emerged as an exciting alternative to produce non-stop parts to overcome this limitation. The continuous FFF is based on a printing process at a variable angle (30°, 45°, or 60°) onto a moving conveyor belt (Figure 5). This strategy has two essential advantages. On the one hand, by printing onto a conveyor belt platform and providing enough print material, there are no longer any limitations on the printed part length along the axis parallel to the belt (Z-axis). The axis aligned with the conveyor belt can be infinitely long, allowing for parts whose size dramatically exceeds that of the printer itself [29]. On the other hand, the moving conveyor belt will enable it to print continuously, which means that the printer can keep printing parts, and they will move down the conveyor belt once they are completed and then collected in a can. In addition, printing layers at 45° or more makes it possible to minimize the need for support material for overhang parts compared to typical FFF [30]. 

The idea of continuous 3DP is not new; FFF printing over a conveyor belt was first mentioned in the RepRap forums in 2008, but in 2016, Bill Stelle fabricated a prototype in conjunction with MakerBot Inc. This product was coined as Automated Build Platform (ABP) but never became a sale-product, not even as an accessory for the MakerBot printers. In the same year, Andreas Bastian, from Autodesk, worked on a similar device, the Lum Printer, that never reached the market. These devices were based on a standard FFF printer with a conveyor belt integrated into the printing platform. The tilting of the printer head (or platform) was included later in 2017 by Brook Drumm in the 3DP device PrintrBelt, the first continuous 3DP on the market. Despite the advances, this technology remains unknown to most 3DP users, possibly due to the lack of resolution due to tilting angles, more significant printing times, or elevated buying prices. These FFF printers are still limited to small-mid scale parts production (from 10 to 1000 objects), which is not tempting for many large companies [30]. Nowadays, few brands sell this type of printer, including the BlackBelt 3D printer from BlackBelt, the Powerbelt3D Zero from Powerbelt, the White Knight 3D printer from NAK 3D designs, and the recently released 3D PrintMill from Creality Ltd. (Shenzhen, China). 

### 3.3. From X-Y Layer-by-Layer to Multiaxial 3D Printing

Some brands have launched FFF desktop printers in the last few years based on a mixture of subtractive and additive processes. These printers were commonly called “All-in-one” 3D printers, boasting laser engraving and cutting, vinyl cutting, drawing, and more. Nowadays, few brands sell this type of printer due to its high cost and complex manipulation. The most remarkable options currently present in the market are model VX from ZMorph, the system 30M from Hyrel 3D, the H-series printer from Diabase, and the 5AXISMAKER CNC from 5AXISWORKS.

However, despite this versatility, extrusion 3D printers are designed to deposit layers in the X-Y plane and then move in the Z-axis to deposit the next layer. This deposition mechanism introduces a degree of anisotropy in the fabricated parts that provoke essential differences in the part behavior depending on the solicitation. Pioneer works proposed by William Yerazunis described alternatives to the traditional deposition method [31]. He describes this technology using the term 5D printing [32], in which the printhead and the object move at 5 different angles, i.e., the extruder moves with three freedom degrees, and the platform can be tilted and thus offering two different axis movements, allowing to print objects with curved layers instead of flat layers. This improvement enables the fabrication of stronger parts with intricate designs. Using the 5D printing concept, several improvements have been reported in the last years, in which the deposition approach has changed from an X-Y plane to a free-form deposition. 

#### 3.3.1. Rotational Axis 3D Printing

Multi-axis 3D printing is based on a conventional FFF printer head that can move along at least one different axis. Typically, the FFF printer head can move in the three common cartesian axes, but some methodologies can add different degrees of freedom to the printing process. We exclude the polar and delta 3DP devices from this section because while they work on non-cartesian axes, they are still three-axial devices. 

An illustrative example of multi-axis 3DP is reported by Wüthrich et al. [33], who developed a novel 4-axis method to print overhangs without support material using a rotative printing head. For this novel printing process, the printhead is rotated 45° around a horizontal axis and equipped with a vertical rotational axis (Figure 6a). The printhead no longer deposits layers parallel to the build platform but moves 45° following a conical surface. With these cone-shaped layers, the printable angles increase by 45°, which leads to printable overhangs of up to approximately 100°. Figure 6b shows two different objects that have been printed as a prototype using different shape-dependent strategies.

#### 3.3.2. Robotic Arm 3D Printing

Another standard methodology for multi-axis 3DP uses an FFF printhead located at the robotic arms’ end. Spatial 3D printing via robotic extrusion offers several advantages over conventional AM methods, both in speed and strength. However, this multi-axis 3DP has many challenges, such as avoiding collision between the robotic arm and the already printed parts or ensuring material self-support during printing. Another multi-axis 3DP challenge limits the starting and stopping points during printing to avoid structural weaknesses [34]. Several studies tried to improve the printing path to reduce printing time and avoid the previously mentioned problems; for example, Huang et al. [35] optimized the methodology to print non-standard, complex, and irregular topologies quickly and flexibly (Figure 7a). These non-standard topologies have considerable potential in design, both for visual and material efficiency effects. They use specialized software for the motion planning framework software, called Choreo, to automatize tedious architectural processes such as assembly sequence, end-effector pose, joint configuration, and transition trajectory (Figure 7b). In Figure 7c, it is possible to observe a superposition of the final 3D-printed non-standard structure together with the optimal printing path determined by Choreo.

Similarly, Piker and Maddock [34] improve the printing path and the meshing parameters to fabricate a stable structure using topological irregular frames that can bear stresses more efficiently. In Figure 8a, it is possible to observe the robotic arm while printing the structure. In Figure 8b, it is possible to perceive the principal stresses within the printed shell, which were determined by using the software Karamba for structural analysis. Accordingly, it was possible to print a structurally stable 2m-high self-standing piece using light materials and multi-axis 3D printing technologies (Figure 8c). 

This technology could also print various types of thermoplastics useful in different applications. For example, Kwon et al. [36] fabricate carbon fiber-reinforced thermoplastic (CFRP) using multi-axis 3DP to create lightweight custom freeform building components at reduced costs. The proposed method eliminates the need for expensive negative molds (moldless fabrication), necessitating minimal scaffolding/support. They fabricate transparent plastic façades for decoration purposes. The internal design was manufactured using CFRP to increase its strength and improve its visual aspect. Figure 9 shows the different steps followed to fabricate the façade structure using thermoplastic materials and CFRP; also, a photograph of the final obtained part is depicted.

The multi-axis robotic arms could also be used as a base for methodologies that involve subtractive-additive processes for object production; for example, Ko et al. [37] utilize this technology to print freeform molds of clay. Multi-axis clay 3D printing was performed using three additive-subtractive methods: hotwire cutter, spindle, and clay extruder (Figure 10a). The first two subtractive methods were used to fabricate the curved molds in expanded polystyrene (EPS), while the third method allows printing with clay over the curved molds of EPS (Figure 10b). It was also possible to fabricate a larger-scale ceramic structure consisting of nineteen panels assembled to create a larger decorative or architectural structure (Figure 10c). 

Mostafavi et al. [38] use this technology to implement a multimode robotic fabrication methodology. They presented a customized automated fabrication workflow composed of subtraction of EPS and manufactured 3D-printed silicone. Integrating these two methods allows for a symbiosis of hard and soft materials. Figure 11a depicts a subtractive process using a robotic arm with a hot wire to cut the EPS chair’s overall form. Figure 11b shows the local milling subtractive process used to fabricate this cell design on the chair. Later, these cavities were filled following the same cell pattern, using a specialized silicone depositing system. The cell-type structures used in the chair prototype design were fabricated by adjusting several parameters for the subtractive and additive processes. Figure 12 shows some of the experiments that optimize the different types of silicone extruding structures, such as linear or cellular printing over a freeform fabric, as a pilot test of the technology. 

### 3.4. Improvements on the Extrusion System

The current extrusion systems employed in FFF and DIW present several limitations that require further improvements. For instance, FFF and DIW use nozzles/needles, respectively, with diameters above 150–200 microns, limiting the printed part’s resolution. Moreover, for FFF, the heating element required to fuse the polymer increases the temperature of adjacent areas for long periods. Finally, printing thermoplastic elastomers with low hardness (below 70 Shore A) is rather complex, if not impossible, since the filament’s elastic nature does not permit a filament-controlled feed. With all the aspects mentioned above in mind, this section’s focus is to summarize the most relevant efforts to overcome those limitations. 

#### 3.4.1. Melt Electrospinning/Solvent Electrospinning

Rivera and Hudson [39] developed an innovative method (melt electrospinning) that mixes two fabrication techniques: electrospinning and FFF 3D printing. The system is based on an electrospinning spinneret connected to a high voltage source, parallel heated to fuse the polymeric material deposited as fibers. The heated spinneret could be transformed into a 3D FFF printing head with a few modifications by incorporating a moving platform and an X-Y translation system over the platform. Using this technology, the shifting from FFF to electrospinning is fast and secure, allowing for fabricating custom-shaped textile sheets (such as wool felt) and rigid plastic material using a single compound (i.e., polylactic acid, PLA in this case). Figure 13a shows a schematic representation in their two different modes: FFF and electrospinning. Figure 13b depict different possible applications of the objects printed with this technology, such as a custom-shaped capacitive sensor formed by electrospinning textile deposited over conductive materials or an origami-style lamp printed using rigid plastic and electrospun textiles. A custom-shaped flower made of electrospun textile and rigid plastic actuates based on the soil’s water level sensed using an electrospun liquid absorption sensor. In this example, when the soil is dry, the flower closes its petals, but the flower’s petals automatically open when the soil is moist. 

##### D Microwave Printing (Charged Materials)

This novel technology is based on standard FFF printing, with the difference that a microwave heater is used to fuse the material inside the printer head. Li et al. [40,41] utilize CFRP to print with this methodology. The filament first goes through an elaborate single-mode coaxial resonant applicator. The microwave is then transferred from the coupling port and transformed into a TEM (transverse electric and magnetic field) wave in the applicator. The TEM’s magnetic field stimulates induced currents in each carbon fiber and generates heating. With the advantage of selective Joule heating, only the CFRP filaments are rapidly heated, while the applicator and surrounding mediums remain at room temperature (Figure 14a).

Moreover, it is also possible to print trusses and complex structures without the use of supports. The 3D microwave printing allows fabricating CFRP components at a much higher speed than traditional 3D printing technologies (Figure 14b). In their work, Li et al. optimize the 3D microwave printing temperature by combining prediction models and a step-proportional-integral-derivative (step-PID) controller to reduce the printing temperature difference of the CFRP filaments during the printing process. The experimental results show accurate temperature control and high tensile strength (close to 360 MPa), 60–80% higher than the conventional FFF printing methods (Figure 14c).

#### 3.4.2. Pellet 3D Printing

Today, many polymeric materials are available for FFF printing, ranging from thermoplastics, composites, and even thermoplastic elastomers. However, biopolymers, material blends, and low-hardness thermoplastic elastomers are examples of materials that are not easy to print. 

This issue is particularly relevant in some research areas where multi-material printing from already printed features (recycled material). Compared with filaments, plastic pellets have a lower cost and straightforward production process [42,43]. Besides, a few additional limitations are related to using new materials in FFF printers, such as the specific size and properties required for the filament. For example, the printing materials should be sufficiently rigid in the FFF technology to withstand the counter-rotating rollers’ force. As the elastomers have less rigidity and low column strength, the existing feeding system of commercial FFF cannot process elastomers’ filaments. When the rollers push the elastomer filament into the FFF machine’s liquefier, it buckles due to low column strength and flexibility. To overcome the limitations mentioned above, the opening of FFF to a vast myriad of novel materials was necessary. Reddy et al. [44] introduced the Extruder Deposition Process (EDP), the initial concept of direct extrusion. They analyzed the effect of different process variables and compared the results obtained with filament 3D printers.

Similarly, Volpato et al. [45] proposed a piston-driven extrusion system using polypropylene to produce a continuous, however defective, extrusion process. Later, on a large scale, Liu et al. [42] introduced the concept of Fused Pellet Modeling (FPM), which combines the layer-by-layer construction systems of conventional 3D printing with screw-based extrusion systems adapted from injection molding tools to additive processes. These adaptations made it possible to increase the nozzle size and layer height and reduce printed parts’ costs due to the lower cost of polymer pellets.

Moreno-Nieto et al. [46] developed two 3D prototypes of significant size for the naval industry (2 m^3^ toilets) using a pellet-based extrusion system as an illustrative example of this methodology’s great potential. PLA and ABS as flame retardants were used to create an object of low cost and reduced weight compared to the original cabin toilets.

### 3.5. Reducing/Avoiding the Use of Supports: Printing in Baths

Another critical issue in FFF printing is using support structures that need to be removed from the final part, provoking additional material requirements during the printing process. Several strategies are under investigation to avoid using supports, including printing in gel baths, by precipitating media, or using bioprinting as a tool for minimally invasive surgery.

#### 3.5.1. Rapid-Liquid Printing (RLP)

AM must face challenges to reach its full potential, such as long printing times, build volumes, and limited material properties [47]. RLP was developed as a technique to address some of these limitations by using a granular gel tank as a reusable support medium. Using RLP, it is possible to significantly increase speed, size, and material properties. This technique allows printing material in any direction without building it layer by layer, thus overpassing the common anisotropy problem of FFF or SLA techniques. For RLP, the raw materials only need to be photopolymerizable or chemically curable. Therefore, the range of available materials expands to rubbers, foams, and high-quality industrial-grade plastics. 

The RLP system consists of three main components: the control platform that allows the system to move in three dimensions, the deposition system (which controls the flow rate, size, and shape of the printed liquid material), and the granular gel tank that it acts as a support medium (Figure 15). The gel acts as a reusable backing material, allowing users to print any shape without additional scaffolding, avoiding any material waste. Once the material is cured, the printed part can be removed, then rinsed, and the remaining gel can be reused.

Another remarkable example was the report by Feinberg et al. [48], who developed a method for extrusion printing within a dissolvable support bath, which comprises a slurry of gelatin microparticles that locks the extruded bioink in 3D space during printing. The technique was coined as a freeform reversible embedding of suspended hydrogels, or FRESH, for simplicity. The bioink, which is conformed mainly by soft proteins and polysaccharides, is embedded in a secondary hydrogel material, which serves as a temporary, thermoreversible, and biocompatible support, allowing hydrated print materials such as alginate, collagen, and fibrin. FRESH bioprinting’s key advance is the support bath preparation, which considerably enhances the printing resolution compared to other bioprinting techniques [49]. Figure 16 shows some of the possible structures and materials which can be printed via this method.

##### 3D Bioprinting and Robotic-Assisted Minimally Invasive Surgery

3D bioprinting is an advanced process that mixes healthcare and AM fields [50]. This technology has progressed in the past few years, presenting innovations, such as tissue and organ fabrication, prosthetics, implants, and pharmaceutical research, such as drug dosage forms and delivery [51]. Bioprinting benefits are extensive and valuable for the customization and personalization of biomedical products [52], including the fabrication of functional tissue or organs. In the US, just 18% of the patients on the waiting list received an organ transplant [53]. With the finality to solve this donor problem, bioprinting has emerged as a technology that allows organ printing over biocompatible matrices using cells from the same subject to minimize organ rejection [54]. 

For example, Yao et al. [55] develop a new type of regenerative medicine that combines 3D bio-printing and robotic-assisted minimally invasive surgery techniques. They investigated Remote Centre of Motion (RCM) feasibility and viscous material extrusion 3D printing, traditionally used in robotic-assisted minimally invasive surgery (MIS). Via the usage of a newly developed RCM mechanism-based robotic system used for touch probe scanning, an osteochondral defect was created by milling and later restoring the articular surface with a 3D printable hydrogel (photocurable alginate-poly (ethylene glycol) diacrylate). Figure 17 shows some schematics and images of the surgery setup.

Recent efforts have also been focused on the in situ and in vivo bioprinting technologies development, the so-called bedside 3D bioprinting. The in situ 3D bioprinters need to incorporate a surface tracking mechanism to print over irregular and mobile surfaces and multi-axis support or robotic arm, which allows the printing over these curve surfaces [56]. Recently, McAlpine et al. [57] developed a tracking surface system to print compliant biomedical devices on live human organs directly. They created an in-situ 3D printing system that estimates the target surface’s motion and deformation to adapt the toolpath in real-time. A hydrogel-based sensor was deposited on a porcine lung under respiration-induced deformation using this printing system. This adaptive 3D printing approach may enhance robot-assisted medical treatments with additive manufacturing capabilities, enabling autonomous and direct printing of wearable electronics and biological materials inside the human body, the first step to the bedside and in vivo 3D bioprinting [58]. In Figure 18, it is possible to observe the procedure followed in carrying out an in-situ 3D bioprinting of deformation sensor over a porcine breathing lung. 

#### 3.5.2. Immersion Precipitation 3D Printing (Ip-3DP)

Researchers at the Soft Fluidics Lab at Singapore University of Technology and Design (SUTD) developed a new 3D printing method to manufacture porous 3D materials in one step, called Immersion Precipitation 3D Printing (Ip-3DP). In this case, the inks (polymers) were printed directly in a bath of non-solvent media, which quickly solidified by precipitation. Spontaneous solidification by immersion precipitation generated porosity at micro to nanoscales, which can be easily controlled by the concentration of polymers and additives in the mixture, as well as the type of solvent (Figure 19). In general, solvent extraction occurs much faster than solvent evaporation. Therefore, the methodology developed allowed a more comprehensive selection of solvents with low vapor pressure (water, DMF, and DMSO) and, at the same time, the use of thermoplastic polymers as ink. 

### 3.6. Multimaterial Parts Prepared by Material Extrusion

FFF technology enables multi-material printing using more than one extruder. In FFF, these filaments are made of a single polymer, polymer blends, or even polymeric composites [60,61]. In addition to this, alternative multi-material structures in which different classes of materials, such as ceramics, metals, polymers, and carbon-based materials, can be combined using direct ink writing (DIW). In an excellent review, Rocha et al. [19] describe the alternatives to take advantage of DIW, i.e., 3DP through the material extrusion of viscoplastic “ink” pastes. This technology creates complex 3D shapes using different materials by formulating a paste with controlled rheological properties (a shear-thinning yield stress fluid) [62,63,64,65,66,67,68].

An essential advantage of DIW, similar to FFF, is that it enables a combination of different formulations into complex structures by using multiple extrusion nozzles (Figure 20). However, so far, only a few examples are considered ‘truly’ multi-material structures due to the restrictions imposed by the post-printing steps, such as drying, debinding, and consolidation, which limit the combinations of materials with different properties (e.g., thermal expansion, melting point, or oxygen sensitivity) [19].

## 4. VAT Photopolymerization

VAT photopolymerization comprises those AM techniques that employ a liquid photosensitive resin as the printing material. This photosensitive resin is placed in a vat, and, depending on the irradiation source and the relative position of the building platform, four variants could be reported so far. VAT photopolymerization is now extensively employed due to the high resolution achieved. Besides, VAT printers’ cost has been continuously decreasing in recent years, reaching a few hundred US dollars.

This section will highlight the recent improvements to overcome the already depicted limitations in terms of fabrication time, the size of the fabricated object, the resolution, and the possibility of fabricating multimaterial parts.

### 4.1. From Step-by-Step Photopolymerization to Continuous 3D Fabrication (CLIP)

SLA—Stereolithography was the first AM technology developed simultaneously in France [69] and the USA [70]. In 1986, 3D Systems was founded by Chuck Hull to commercialize this technology. Photolithographic systems build shapes using light to solidify photosensitive resins selectively. A liquid photopolymer in a vat is selectively and spatially cured by light-activated polymerization in SLA technology, generally using UV light from a laser source. Digital Light Processing (DLP) is a technology derived from SLA. In this case, the UV source is a digital projector instead of a UV laser. Unlike SLA, each layer is exposed not point-by-point but rather all-at-once with a selectively masked light [15].

However, both SLA and DLP involve a discontinuous step-by-step process. After the UV light projection or the laser exposure, the surface is cleaned to regenerate the oxygen layer to prevent the part’s adhesion to the vat’s bottom. DeSimone et al. [71] developed and patented a new method to continuously fabricate the 3D printing part to reduce the fabrication time; the technique was called Continuous Liquid Interface Production (CLIP) [72]. DeSimone’s technology is similar to SLA or DLP printing but has a crucial difference; CLIP technology possesses an oxygen-permeable film at the bottom of the vat resin to locally inhibit polymerization (Figure 21a). DeSimone et al. [73] show how controlled oxygen inhibition can be used to enable a more straightforward and faster SLA. Typically, oxygen inhibition leads to incomplete cure and surface tackiness. Still, when SLA/DLP is conducted above an oxygen-permeable build window, CLIP is enabled by creating an oxygen-containing “dead zone”, which is a thin uncured liquid layer between the window and the cured part surface. The presence of this dead zone allows printing 3D parts from liquid resin continuously, without the wetting/dewetting process between layers necessary for typical SLA or DLP techniques. The CLIP method saves considerable time printing 3D parts, manufacturing objects 5 to 10 times faster, without losing resolution [73]. Some printed parts obtained with this method are depicted in Figure 21b, together with SEM micrographs of ramp test patterns produced at the same print speed regardless of 3D model slicing thickness (Figure 21c). Some further advances performed by the same group allow controlling the dead zone’s stability via electrochemical methods [74]. Interestingly, the mechanical properties of the 3D parts formed via this method (with the smaller resolution) do not show any geometrical anisotropy [75], thus not presenting the common problem with layer-by-layer 3DP.

However, the group of DeSimone et al. was not the only one investigating and developing advances in the CLIP printing method. For example, Qi et al. [76] use this technology to fabricate epoxy thermosets materials in a two-step polymerization process. Most 3DP thermosetting polymers suffer from inferior mechanical properties and low printing speed. Still, by using CLIP, these problems could be easily solved by including a photocurable and a thermocurable epoxy resin. After printing, the part is thermally cured at elevated temperature, thus producing an interpenetrating network between the photopolymerized structure, previously formed via CLIP, and the new thermopolymerized polymeric network (Figure 22). This process greatly enhances the mechanical properties of the material. The printing speed was accelerated to almost 220 mm h^−1^, more than seven times faster than the SLA/DLP printing process. 

### 4.2. Fast Printing and Large Sizes

Another critical challenge in VAT polymerization is the part dimensions. Size is not essential for many VAT applications, such as jewelry or dental. However, with the development of novel materials such as elastomeric resins or thermosets with mechanical properties similar to polypropylene, novel applications are envisaged, requiring larger parts in some cases.

#### 4.2.1. High Area Rapid Printing (HARP)

In this context, Walker and Hedrick et al. [77] have reported a novel approach based on VAT photopolymerization to fabricate larger parts called High Area Rapid Printing (HARP). The HARP process is based on SLA technology, and it has continuous printing over a large area and rapid vertical print speeds. Azul 3D launched its first 3D printer featuring HARP technologies, which can print objects up to 4 m tall with a speed of ~0.5 m h^−1^. The printer operates on a UV-curable resin that floats on an immiscible liquid (fluorinated oil), favoring the heat remotion and preventing adhesion to the bed. Aizenberg et al. [78] focused their research on creating surfaces with variable hydrophilic/hydrophobic balances. Accordingly, the authors report a synthetic liquid-repellent surface named slippery liquid-infused porous surface (SLIPS). The lubricant fluids used for the experiments were perfluorinated (3M Fluoroinert FC-70, DuPont Krytox 100 and 103) with two kinds of porous substrate (ordered epoxy-resin-based nanostructured and a random network of Teflon nanofibrous membranes). Lubricant fluids were added to the porous substrates to prepare the SLIPS, which can serve as omniophobic materials capable of meeting emerging needs in biomedical fluid handling, fuel transport, anti-fouling, anti-icing, self-cleaning windows, optical devices, and many more areas that are beyond the reach of current technologies [78].

HARP utilized a fluorinated phase found in constant motion and filtered to remove the microparticles solids generated during the SLA printing process [79]. This compound decreases the adhesion force (i.e., static versus dynamic) and, in turn, generates a slip boundary in the solid-liquid interface (Figure 23a). Finally, this technique does not require a dead oxygen layer because it is compatible with oxygen-sensitive and -insensitive ink chemistries. Figure 23b shows some proof-of-concept structures made from hard plastics, ceramic precursors, and elastomers using HARP technology, which would not have been possible using oxygen-dependent DLP technologies [77]. 

#### 4.2.2. Computed Axial Lithography (CAL)

Although 3DP refers to the fabrication of objects in three dimensions, this method corresponds to a continuous assembly of several 2D printed films layer-by-layer. This assembling process dramatically increases the fabrication time required. Indeed, volumetric printing is not easy to achieve because the simultaneous formation of the printed part’s whole volume as a unit operation is one of the last remaining barriers to overcome for rapid 3D part fabrication spanning all three spatial dimensions, with no substrate or support structures required. Shusteff et al. [80,81] modify the DLP methodology by adding rotation to the photocurable resin tank (Figure 24a), allowing photopolymerizing simultaneously in the printed part from multiple beams projected. The superposition of patterned optical fields into the photosensitive resin enables the production of volumetric 3D structures in reduced fabrication times successfully. This novel methodology was coined as computed axial lithography (CAL). Shusteff et al. implement this approach using holographic patterning of light fields, demonstrating the fabrication of various parts with complex internal structures. The main advance performed by Shusteff et al. is to introduce molecular oxygen (O_2_) dissolved in the resin (or another polymerization-inhibiting species mixed into the formulation) to provide the non-linearity necessary for “threshold” behavior in the polymerization process and thus control the volumetric polymerization of the resin. 

Later, in 2019 [82], the same group optimized this technology, producing high-resolution printed structures with complex inner and outer geometries of centimeter-scale dimensions in reducing printing times (from 30 to 120 s). The fabrication of support-free structures was also accomplished, together with soft material printing. Figure 24a,b show a schematic description of the CAL process and the device used to perform it. Figure 24c shows a series of photographs of the building process in a CAL printer. The final result was obtained with four different materials to print a miniature reproduction of “the thinker” in less than a minute. 

### 4.3. Improving Resolution (Micro-SLA)

Wu and Song et al. [83] proposed a one-droplet 3D printing strategy to fabricate controllable 3D structures; the method was called continuous single droplet 3DP. Figure 25 shows a schematic diagram and photographs of the proposed printing process. Thus, a single droplet of liquid resin is deposited on the curing interface (step I), then, the supporting plate makes contact with the center of the resin droplet (step II), the UV-light exposure solidifies the resin layer-by-layer, forming the final 3D structure (steps III and IV). This research uses three different substrates, i.e., fluorinated quartz (F-quartz), a candle soot-based, and a lubricant-infused PDMS slippery substrate (S-PDMS), to investigate the influence of curing interface properties on the one-droplet 3D printing process, concluding that S-PDMS surface is the best choice for one-droplet 3D printing.

The micro-SLA (micro-stereolithography) is a process capable of fabricating 3D microparts using modified techniques of the conventional lithographic principles [84]. The micro-SLA is divided into direct laser writing and mask projection micro-SLA.

Mask projection micro-SLA (PμSL) is a method based on SLA capable of fabricating high-resolution structures [85]. Like conventional SLA, parts are manufactured layer-by-layer via selectively curing the printing area. Stampfl et al. [86] mounted the micro-SLA system on an optical table with an optical setup comprised of a laser source, an acousto-optic modulator, high-precision translational stages, and a processing chamber. The system counts with a compensation mechanism that moves the resin trough in the opposite direction during printing to keep the polymer level constant. Two different materials were printed as tests, organically modified ceramics (ORMOCER) and organic acrylate-based resins. Some of the results obtained via this methodology can be observed in Figure 26. 

Nowadays, few brands sell similar technologies; for example, Nanofabrica Co. sells the model Tera 250, which allows printing “huge parts” (50 × 50 × 100 mm) with high resolution (1 μm) with ABS and ceramic loaded materials. Boston Micro Fabrication (BMF), and Asiga with the model MAX and MAX UV, are other brands that commercialize this type of DLP printer with high resolution [87,88].

### 4.4. Strategies to Fabricate Multimaterial or Intricate Structures by VAT Printing

Like FFF, the preparation of multimaterial parts with intricate structures by VAT photopolymerization still needs further investigation. However, some exciting reports have demonstrated that it is possible to obtain such complex structures.

#### 4.4.1. Hierarchical Intricate Structures

Freeze-drying DLP was recently proposed by Koh et al. [89] as an innovative method based on DLP to fabricate hierarchical porous ceramic structures using a photocurable ceramic slurry that contains a freezing vehicle (frozen camphene-camphor alloy network). The slurry monomer/ceramic mixture was prepared by ball-milling at a proper temperature (70 °C) and deposited as a thin film over a building platform using a recoater. After a short time, this mixture can become rigid due to freezing. This freeze-cast layer can be photopolymerized using a custom-built DLP machine. Finally, freeze-drying frozen camphene-camphor networks can be removed, resulting in micropores of ceramic frameworks surrounded by straight macrochannels (Figure 27). With the finality of examining the effect of freezing vehicle content on micropores distribution, various photocurable ceramics slurries with different freezing vehicle content (40, 50, and 60 vol%) were prepared. Thus, the fraction and size of the micropores increased notably with the freezing vehicle content. At the same time, the compressive strength and modulus decreased, a behavior that could be attributed to an increase in the porosity of the material. 

The same group fabricated macroporous gyroid structures for cell scaffold purposes using freeze-drying DLP technology [90]. In general, all the 3D parts obtained with different camphene-camphor content (40 to 60 vol%) displayed similar cellular response; however, the microporous calcium phosphate (CaP) framework obtained using the highest camphene-camphor content (60 vol%) showed significantly higher cell viability than the obtained using the lowest content (40 vol%). Additionally, the water penetration and in vivo cell forming ability increase with the camphene-camphor content, indicating that these processes are mainly dominated by the high microporosity and pore interconnectivity of the 3D-printed part. These findings suggest the great usefulness of hierarchically porous CaP scaffolds with microporous frameworks for bone scaffold applications. In 2020, the same research group examined the utility of photocurable ceramic/monomer feedstock containing terpene crystals as sublimable porogens for UV curing-assisted 3D plotting to construct ceramic structures comprising microporous filaments [91]. In this case, filaments’ porosity was tailored by adjusting terpene content (50, 60, and 70 vol%) in biphasic calcium phosphate (BCP) feedstocks. Additionally, the compressive strengths of dual-scale porous BCP scaffolds were characterized to evaluate their potential application as bone scaffolds, suggesting that the pores formed in BCP filaments would stimulate blood delivery. 

#### 4.4.2. Multimaterial Structures Based on SLA Technologies

Several research groups recently developed an interesting micro-SLA technology system to fabricate multimaterial structures in reduced times. Lee et al. [92] manufacture a micro-SLA apparatus in an enclosed fluid cell that includes a pumping system to recirculate the liquid resin in the vat, quickly exchanging the printing material without interrupting the process enabling high-resolution multimaterial 3D printing (Figure 28a,b). In Figure 28c is possible to observe some of the parts obtained by employing this multimaterial micro-SLA printing system. 

#### 4.4.3. Direct Laser Writing (DLW)

Direct laser writing (DLW) is a 3D manufacturing technology that offers vast architectural control at submicron scales. DLW technology involves using a tightly focused femtosecond pulsed IR laser to initiate photopolymerization via two-photon (or multi-photon) absorption phenomena at designed locations within a liquid-phase photoreactive material. By positioning the laser, 3D structures comprised of cured photo-material can be additively manufactured with resolutions on the order of 100 nm. Several groups have studied that DLW can be employed to print multi-material systems in which each material corresponds to distinct chemical, biological or optical properties. Sochol et al. [93] reported a facile multi-material DLW system that combines standard PDMS micromolding, impermanent PDMS-to-glass bonding, vacuum-based microfluidic infusion, and in situ DLW techniques to fabricate 3D multi-material microstructures. Figure 29 shows some of the results obtained via multimaterial DLW 3D printing.

## 5. Recent Innovation on Selective Sintering Technologies

The previous sections reviewed the innovations performed in the two most common AM technologies, i.e., material extrusion and vat photopolymerization. This section presents a brief discussion about the third most commercialized AM technology (Figure 2).

The powder bed fusion (PBF) method refers to the selective consolidation of dust particles into 3D objects, using a focused heat, a laser source, or an IR lamp [94]. The most extensively employed approach is selective laser sintering (SLS). In the SLS printing process, 3D objects are built layer-by-layer via sintering of thermoplastic powders through thermal energy resulting from the combination of the increase in temperature and a light source [95]. However, significant aspects that are still to be improved include the resolution, the possibility of producing 3D-printed parts continuously, or the elaboration of multimaterial 3D-printed parts. This section is devoted to the most recent advances reported to improve these issues.

### 5.1. HLS, SLS and Multijet 3D Printing

Multi Jet Fusion (MJF), emerging in 2014, is a Hewlett-Packard (HP) Inc. technology, and, in contrast to SLS, which uses a laser as the heat source, MJF utilizes an array of infrared lamps to fuse the area of interest. This area was previously jetted with a fusing agent that can absorb infrared radiation. The fusing agent is deposited by inkjet nozzles installed in a carriage to the powder bed’s designated regions on the voxel level. Meanwhile, a water-based detailing agent is jetted around the contours of the printed parts to inhibit the fusion of powder near the part edges and improve part resolution. When printing large parts, the detailing agent is also jetted into specific areas within the large pieces to prevent partially excessive thermal accumulation [96]. 

In a recent study, Cai et al. [96] compared both technologies using polyamide 12 (PA12) by evaluating the printed parts’ mechanical properties and printing characteristics. They found anisotropy in tensile and flexural properties in printed specimens of both 3D printers. The mechanical strength tendency in the build orientations for both printed specimens differed, especially in the Z orientation. The tensile strength of the MJF and SLS printed samples in the X and Y orientations was almost identical. However, the tensile strength of the MJF specimens in the Z orientation was ~25% higher than their SLS counterparts. MJF parts had a better surface finish than the SLS specimens, except the top surface.

### 5.2. Continuous SLS 3D Printing

In SLS 3D printing, conventional production is characterized by two different steps that are repeated cyclically. In the first cycle, the platform moves downwards, forming the material layer-by-layer via thermal action, repeating this process until it reaches the configuration’s maximum height. The machine stops, and the working box is removed, containing the particulate support material and the printed product. To avoid this problem, Günther et al. [97] described a 3D printing machine’s configuration as a continuous SLS 3D printer. This configuration allows the structures to be unpacked during the current printing process. The authors overcame SLS’s limitations by creating a printer model that allows extrudate movement through a horizontal conveyor belt. In this case, the workbox’s lower wall is replaced by a conveyor belt arranged at an angle that allows the parts to move. This type of machine could be used instead of conventional core production machines to achieve fast manufacturing, due to the absence of workboxes and a simplified single-use feeding system, being able to produce a large batch size of small parts, as well as large parts in the dimension of the machine-building space [97].

### 5.3. Multimaterial Parts Fabricated by Selective Laser Sintering

Finally, a critical limitation in SLS is related to combining different materials. The properties of the 3D-printed part have been modulated using the appropriate material in each case. For example, the modulus of the object can be finely tuned by introducing more material, i.e., making the part more solid (reducing porosity increases the final Shore) or reducing the amount of material employed by introducing cavities in the design. However, combining more than one material in the same printed part remains a challenging task. 

Some strategies have focused on developing multi-material AM methods for laser sintering equipment, either by placing an initial layer with spaces for material to be filled or replacing un-sintered material in each layer with a secondary powder using a vacuum. However, both methodologies have evidenced some limitations in terms of contamination with un-sintered material [98].

Attempting to improve these limitations, other alternatives include Laser Engineered Net Shaping (LENS) together with Direct Energy Deposition (DED) processes, which jet multiple powders into the focal point of a laser beam [99]. The LENS process has been employed to fabricate metallic components and presents several advantages, including the ability to add material to existing parts enabling the fabrication of complex geometries. However, this approach is generally wasteful in powder usage since the powders cannot be easily separated or retrieved after the spraying. In this context, more recently, Whitehead and Lipson [100] reported an SLS process design to sinter material microparticles in which the laser is directed vertically upwards into a thin layer of powder through a borosilicate glass pane underneath the print bed. They call this process Inverted Laser Sintering (ILS). The first step of ILS involves coating the top face of the glass surface with a release agent, and on top of this release agent, the powder evenly uses vibration to form a monolayer. The excess material can be removed using a vacuum device, as the release agent captures a single layer of powder. A substrate is pressed on top of the unfused powder monolayer, and a blue laser is used to fuse the particulate material onto this substrate selectively. Finally, the substrate is lifted, and the material on the powder glass plate is replenished.

Using multiple glass plates in parallel, it is possible to use different materials, thereby enabling the manufacture of graded and multi-material parts. Multi-material fabrication is accomplished by transporting the substrate between separate glass supports for each material, thus preventing particle mixing. Multiple glass beds with identical material could also be used parallel to accelerate the processing so that one bed is being prepared while the other is being used. Transporting the part between multiple print beds allows for an integrated cleaning mechanism that removes loose powder to prevent cross-contamination. This printing method would also reduce the amount of material needed to generate a part by eliminating the need for the surrounding passive support material bed, thus reducing the amount of material required to be exposed to a heated environment to generate the print. 

## 6. Advances in the Design of Novel Materials for AM

In addition to the technological advances described in the previous sections related to the most widely employed 3D printing technologies (FFF and SLA), in parallel, a considerable effort has been carried out to develop novel materials to enlarge the range of applications of the 3D-printed parts. This section is not intended to provide an exhaustive overview of the polymeric materials employed in 3D printing but instead focus on some selected examples. The readers should refer to other more specialized reviews for a general overview of polymers and additive manufacturing [15,101].

### 6.1. Liquid Elastomer Printing

Brun et al. [102] designed a straightforward methodology to functionalize soft elastomers by printing drops of water at the surface, giving shape and function to the elastic matrix. The printed drop patterns can be used to program the deformation of thin elastomeric structures by controlling swelling, in addition to being used as micro-reservoirs, transporting, and protecting small amounts of liquid, which can be released when piercing the capsule membrane. 

After the elastomer crosslink, the authors demonstrate that the injection-printed drops remain encapsulated, forming millimeter cavities (protected by a thin elastomer membrane at the end of the process) whose position and geometry can be adapted. This work offers an efficient route to manufacture structures where liquid inclusions are used, significantly impacting practical applications such as compartmentalized reactors, drug administration, biocompatible materials with scaffolds, and encapsulated active matter.

### 6.2. Reinforced Polymers for Additive Manufacturing

As has been already highlighted, reinforcing polymeric materials with inorganic fibers or particles is a commonly employed strategy to improve their performance. 3DP of reinforced materials has already been extensively described elsewhere. However, for the case of reinforced materials with fibers, a significant limitation is related to the fibers’ damage during the different preprocessing steps, i.e., filament fabrication and 3D printing process. In this sense, it is fascinating to print polymers reinforced with continuous carbon fiber. For instance, Van der Klift et al., in 2016, studied the production capabilities of the Mark One^®^3D for printing different types of carbon fiber reinforced thermoplastic (CFRT) [103]. Matsuzaki et al. [104] developed a method to print CFRT based on FFF in the same year. Here, PLA was used as the matrix, while carbon fibers, or twisted yarns of natural jute, were used as the reinforcements. As a result, the reinforced with unidirectional carbon fiber showed better mechanical properties than the jute-reinforced and unreinforced thermoplastics. 

Another illustrative example of CFRT printing was reported by Li et al. [105]. Their article proposes a prototyping approach for the rapid and continuous printing of CFR-PLA. The novel extrusion nozzle and path control methods were designed to print curve surfaces using a polymeric blend between carbon fiber and PLA resin. The schematics of the designed extrusion device are shown in Figure 30a. Straight and curve paths for 3D printing of CFRP can be effectively achieved. Carbon fibers’ preprocessing improved the interfacial strength, considering the weak bonding interface between carbon fiber and PLA. Three different models were tested using the proposed methodology (unidirectional flat part, hollow-out aero foil, and a circle, Figure 30b). The results indicated that the modified CFRT present tensile and flexural strengths much higher than the original parts. The modified CFRT samples’ storage modulus was higher than the PLA and the original fiber reinforced samples for 166% and 351%, respectively. The SEM images indicated that the preferable bonding interfaces were achieved by modifying the CFRT composites. This rapid prototyping technology for the continuous carbon fiber composite can manufacture complex and high-performance composite parts, especially for complex aircraft structures [106].

A second alternative to producing composites with minor damage to the fibers was reported by Hollins et al. [107], who developed a process known as localized in-plane thermal assisted 3D printing (LITA). This process uses thermoset polymers with carbon fiber to manufacture a composite with good mechanical properties, high thermal stability, design flexibility, low cost, reliability, and repeatability. Carbon fibers are strategically designed with spaces or pores for absorbing a liquid polymer. After this, the fibers are heated, allowing the formation of a 3D-printed structure. 

The technique is based on a continuous capillary effect, which results from a thermal gradient in movement along the carbon fiber surfaces, facilitating the flow of liquid polymer in space, acquiring the shape of a tube between neighboring carbon fibers. Afterward, it performs the polymeric resin curing from the heated fiber surfaces to the surrounding space. Then, the liquid resin moves towards the carbon fibers’ higher temperature regions, filling empty spaces. The printing mechanism is composed of a printing head that contains carbon fibers, a Joule heater, a resin distributor, and a robotic arm responsible for the 3D vement of the printing head (Figure 31a,b). 

There is a drawback concerning this type of reinforced polymeric material. Sometimes, there is no good compatibility between the polymer and the reinforcement, thus reducing the material properties to be obtained [108,109,110]. The use of additives to improve material properties for processing and application is common practice in sheet molding compounds for traditional injection molding applications [111,112]. Additives can be used to modify melt flow [113], increase strength, and/or decrease warpage [114]. Fillers have also recently been incorporated in FFF filaments to alter material properties, such as shrinkage/warpage [115], rheology, or add functionality, such as magnetic properties [116].

### 6.3. High-Temperature Materials

The most extensively employed materials for FFF are commodity thermoplastics, including PLA or ABS, and thermoplastic polyurethane, such as TPU. Engineering polymers and high-performance polymers have been less explored, mainly due to technical difficulties in the printing process. However, the advances in 3DP and the arrival of modern high-performance thermoplastic polymers, such as polyether ether ketone (PEEK), polyphenylene sulfide (PPS), polysulfone (PSU), opens unprecedented possibilities for successful manufacturing of high-performance engineering and biomedical devices [117]. PEEK has been widely used in aeronautical and biomedical applications. For example, Berreta et al. reported 3D cranial implants’ fabrication of biocompatible unfilled PEEK [118,119,120]. However, the main difficulty in fabricating PEEK parts using AM technology is the high melting temperature of the material (nearly 400 °C), making parts undergo significant temperature change, producing considerable internal stress, warpages and delamination during the printing process. Several efforts have been devoted to mitigating this problem [121]. Arif et al. [122] examined three different configurations, and they found that specimens built vertically were more prone to delamination, exhibiting low mechanical performance due to high thermal gradient along the build direction. Minimizing thermal gradients across beads is the key to producing parts with excellent macroscopic properties. Wang et al. reported a new type of extrusion-type printing nozzle for rapid prototyping of PEEK materials. They could form stable parts from PEEK by using a screw extrusion method and designing an exchangeable printing head with two different types of nozzles (line and plane printing nozzles) [123]. 

In this context, technological improvements have allowed the commercialization of novel 3D printers from different companies, including Intamsys, Apium, 3D Genze, Stratasys, and Zortrax. These 3D printers allowed maximal extruder temperatures of around 500 °C. In addition, they permit control over the bed temperature (in the range of 200–300 °C) and the chamber (up to 250 °C). These two issues are critical to assuring the adhesion of the 3D-printed part to the bed and reducing warping produced by crystallization in the layer-by-layer process.

### 6.4. Fabrication of Low-Cost Metallic and Hybrid Metallic-Polymeric Parts

One of the main issues related to thermoplastic 3DP is that the material itself does not fulfill the mechanical, thermal, or electrical requirements demanded by several industrial applications [124]; this is why there is a growing interest in developing an efficient method to carry out metal 3DP. While 3DP with thermoplastic materials is highly advanced and can readily create complex geometries at low cost and relatively short times, 3D printing of metals is still challenging due to its price. Bulk metallic glasses (BMGs) are a family of metallic materials that present a supercooled liquid region and a continuous softening behavior upon heating, analogous to thermoplastics. Schroers et al. [125] demonstrate that BMGs are also amenable to extrusion-based 3DP through FFF methodologies. Figure 32a shows a schematic representation of the design used to print metal parts based on FFF technology. Figure 32b shows a photograph of the physical setup of the BMG printer, and Figure 32c depicts a picture of the obtained metallic parts printed via this method. 

Similarly, Oh et al. [126] developed a 3DP system based on FFF to create metallic patterns and parts. Their article investigates several parameters related to the processing procedure and the nozzle’s optimal design to print the metallic materials. A numerical heat transfer simulation was conducted to design the nozzle system; based on the results, a metal 3D printing system with X, Y, and Z stages was constructed. In general, three different types of pattern printing were detected (bulged, uniform, and dashed lines). It was possible to obtain uniform and homogenous printing patterns by altering the printing parameters. 

Recently, in 2020, Liu et al. [127] proposed a novel method to fabricate 3D-printed parts based on metallic materials. This technique was coined as Fused Deposition Modeling and Sintering (FDMS) and is based on FFF printing of a metal/polymer composite filament. Figure 33 shows a schematic illustration of the FDMS process. Firstly, the Green Parts are printed from metal/polymer composite filament by FFF, during which the polymer is melted as the binder, but the metal particles remain solid. Later, Brown Parts was obtained by subjecting the Green Parts to a debinding process to remove most of the polymer binder. The rest polymer binder in the Brown Parts can avoid spreading the metal particles and preserve the parts’ shape. Finally, the Brown Part is sintered to fuse the metal particles to form dense FDMS parts. The materials chosen to fabricate the filament was stainless steel 316L as microparticles (30–50 μm) spread into a polymer matrix of polyformaldehyde (POM) and additives such as polypropylene (PP), dioctyl phthalate (DOP), dibutyl phthalate (DBP), and zinc oxide (ZnO) to increase the fluidity, plasticity, and thermo-stability of the composite. Different microstructural characteristics of the 316L/POM filament were measured, such as the hardness, tensile properties, relative density, and part shrinkage.

The current manufacturing techniques for metal-polymer layered parts usually display long processing cycles to cure the thermoset-based resin, such as epoxy-based fiber-metal laminates (FML). AM is an alternative for automating the FML manufacture, increasing the freedom in intricate part design [128]. The AddJoining technique is an AM method based on the fabrication of layered metal-polymer hybrid structures that combine the principles of AM and materials joining methodologies. The group of Amancio-Filho et al. [129] developed this technique and tested its feasibility by printing aluminum 2024-T3/ABS and aluminum 2024-T3/unreinforced polyamide 6 (PA6)/carbon- fiber-reinforced polyamide 6 (CF-PA6) materials combinations. 

The AddJoining process could be divided into four consecutive steps (Figure 34a): (1)The metallic substrate is placed on a building platform (Figure 34a, top-left).(2)A polymer layer is deposited on a metallic substrate. (Figure 34a, top-right). (3)The subsequent polymer layers are deposited until the desired thickness and sequence of the polymeric part is achieved (Figure 34a, bottom-left).(4)Finally, the metal-polymer layered joint is removed from the building platform (Figure 34a, bottom-right). 

The results demonstrate that the joining process is possible using AM technologies; Figure 34b shows the joints’ cross-sectional microstructure. Direct contact between the coating layer and the aluminum surface could be achieved for both studies. No bond line could be detected between the deposited polymer and coating layers, suggesting intermolecular diffusion and a strong bond formation at the interfaces. However, voids were detected between the CF-PA6 and PA6 layers (marked with white arrows in Figure 34b, right).

In 2019, the same research group recently optimized some printing parameters of the proposed methodology, such as printing temperature, layer thickness, deposition speed, and printing direction, among others. They also investigated the joints’ microstructure and the specimens’ fracture morphology after the mechanical test was performed [130,131]. These results indicate that a proper mechanical interlocking was achieved between the coated metal substrate and the 3D-printed polymer.

### 6.5. Ceramic Parts (Solvent-Cast 3D Printing (SC3DP))

Solvent-cast 3D printing (SC3DP) is another technique based on ink extrusion. These contain metallic particles or any other powder type, together with a binder system (polymer compound, volatile solvent, or additive). The polymer is previously dissolved in a specific solvent and subsequently extruded through a needle onto a collecting surface to form the final material. As the solvent evaporates, the solid polymer remains in the printed structure [132,133]. SC3DP offers multiple benefits, including (i) 3D printing under ambient conditions, (ii) easy adjustment of ink components, (iii) low investment in equipment, and (iv) the potential to fabricate complex structures with hierarchical pores. 

Dong et al. [134] used this methodology (Figure 35a) to manufacture porous, and biodegradable Mg scaffolds arranged topologically. They prepared an ink loaded with an Mg powder with the desired rheological properties, then an SC3DP of the ink was made to form scaffolds with different angles between layers. Finally, the debinding and sintering process was carried out to eliminate the ink’s binder and obtain Mg particles bonded by sintering in a liquid phase. The manufactured material demonstrated the magnesium particles’ successful binding, forming a microporous structure (Figure 35b).

### 6.6. Smart Materials: From 3D to 4D Printing

Different research groups have focused their efforts on designing and preparing smart materials, i.e., materials that can respond to a particular stimulus, thus enabling shape changes in 3D-printed parts. Thus, 3D printing has led to a new field known as 4D printing, whose fourth dimension is time. Skylar Tibbit first introduced this technology in collaboration with Stratasys. The authors reported a printed material programmed to change over time in response to an external stimulus, such as swelling [135]. This pioneering example has served as starting point for various developments in which different smart materials have been employed [15,135]. We are limiting our discussion to polymers, smart materials based on this type of polymeric composites include shape memory polymers (SMPs), hydrogels, and shape memory composites (SMCs), since these three are the most extensively employed. Ryan et al. [135] described that these materials could switch from a temporary state to a stable one. More interestingly, as depicted in Figure 36, this switching behavior can be induced by exposures to changes in electromagnetic radiation, moisture, pH levels, and electrical and magnetic fields.

It is worth mentioning that single and multi-material objects can be straightforwardly fabricated by AM and designed to react to different stimuli (Figure 37 top). In some cases, shape changes can be achieved by using multi-stimuli arrangements. A concept design based on this methodology was proposed by Khare et al. [136], in which an artificial insect constructed from multiple smart materials is illustrated (Figure 37 bottom). This ambitious complex design involves several shape types and changes provided by different multi-stimuli combinations to simultaneously achieve expansion, flexibility, shrinking, or morphing, with the finality to fulfill a particular desired application. 

While it is true that, due to the current development phase of 4D printing technology, the variety of stimuli-responsive materials and design concepts is somewhat limited and thus requires further development to achieve complex architectures.

An interesting example about the potential of 4D printings was reported by Gladman et al. [137], who fabricated a composite hydrogel ink that mimics plant cell walls (Figure 38). It consists of a soft acrylamide matrix reinforced with cellulose fibrils. The composite is printed using a viscoelastic ink composed of an aqueous solution of *N*, *N*-dimethylacrylamide, Irgacure 2959 as photoinitiator, nanoclay, glucose oxidase, glucose, and nanofibrillated cellulose (NFC). The clay particles were used to alter the rheologic and viscoelastic properties necessary to obtain the desirable ink for printing. More significant amounts of clay lead to higher crosslink densities and lower swelling ratios. Glucose oxidase and glucose scavenge the surrounding oxygen, reducing oxygen inhibition during the UV curing. The shape-shifting state of the material described above is irreversible. To achieve reversible shape-shifting behavior in hot and cold water, the poly(*N*, *N*-dimethylacrylamide) needs to be replaced with a thermo-responsive polymer *N*-isopropylacrylamide.

### 6.7. Sustainable Materials for Additive Manufacturing

As has been largely described, AM enables the fabrication of innumerable 3D geometries that other means cannot efficiently produce. However, despite the great promise of AM as an advanced form of future manufacturing, there are still fundamental challenges concerning sustainability that need to be addressed. In this context, there are still material needs for AM that involve sustainable sources of printing inks, resins, and filaments, as well as pathways for polymer recycling, upcycling, and chemical circularity. Sanchez-Rexach et al. [138] reported a complete review about the combination of bio-sourced and biodegradable polymers with AM capabilities to fabricate objects that can be recycled back into feedstock or degraded into nontoxic products after they have served their function. The authors gathered the recent literature on the design and chemistry of the polymers that enable sustainability within the field of AM, with a particular focus on biodegradable and bio-sourced polymers. They also discuss some sustainability-related applications that have emerged because of AM technologies development.

Naturally occurring biopolymers (such as DNA, proteins, and polysaccharides) possess a high molecular weight, translating into inherently viscous polymer solutions. As a result, the processing and printing of these biopolymers in AM processes can be challenging. Some of these biopolymers also require chemical modification to undergo light-initiated cross-linking. Alternatively, synthetic polymers can offer greater control over polymer composition, molecular weight, and polymer architecture to accommodate the printing technique’s requirements. Examples of biopolymers and synthetic polymers for AM are summarized in Figure 39.

A particular case in the design of sustainable materials to be employed in AM is the case of thermosets, commonly used in SLA or DLP technologies. In contrast to thermoplastics, thermosets are inherently non-recyclable because covalent bonds permanently crosslink the polymer chains. As a result, this polymer family is more resistant to solvents and possesses superior thermomechanical properties [138]. 

Covalent adaptive networks (CANs) are polymer networks that contain exchangeable covalent bonds [139,140,141]. A subcategory of CANs, known as vitrimers, is particularly attractive as reprocessable and recyclable materials for AM. Zhang et al. [142] reported an illustrative example of these materials. The authors described an innovative method for preparing a reprocessable thermoset for UV curing-based high-resolution 3D printing. A polymer was produced by employing a photoinitiator and a cross-linker with hydroxy-3-phenoxypropyl acrylate as the monomer, which was produced containing both permanent and dynamic covalent bonds. This capacity allowed the material to be reshaped at an elevated temperature due to the bond-exchange reactions. The same material also demonstrated self-healing properties. After being damaged, the structure was polished, and additional material was added to rebuild the same structure with no mechanical performance losses observed in the previously damaged region. Finally, the material was mechanically reprocessed by grinding the printed structure. Due to the bond-exchange reactions, the resultant powder was subjected to high temperatures to obtain a new ink.

## 7. Conclusions and Futures Perspectives

Material extrusion and VAT photopolymerization are, by far, the most extensively employed AM technologies. These technologies are equally expected to maintain and even increase their impact on today’s market distribution. The reason behind this continuous increase in machine sales is without any doubt related to the affordable price (in particular for material extrusion and vat photopolymerization), but also to the recent developments that permit today the fabrication (just plug-and-play procedures) using a large variety of different materials: rigid, soft, temperature resistant or biocompatible, to mention a few of them.

Despite the advancements, these technologies still have some drawbacks that require consideration. This review addresses the most relevant limitations and analyzes the solutions reported in the recent literature and available in the market to solve or at least minimize them. The improvements in fabrication speed, the alternatives to continuously produce parts, or the increase in the parts’ dimensions and the resolution are currently the center of multiple investigations.

Technological advances in AM also require the research of novel materials adapted for each technology. In effect, a precise material is preferable rather than adapting available materials. In this sense, this review also presents illustrative examples of reinforced polymers for additive manufacturing, the use of high-temperature materials, or the fabrication of low-cost metallic and ceramic parts. Besides, smart materials to elaborate on shape-changing parts or sustainable materials have been discussed.

Ongoing research from both technological and materials points of view will enable further incorporation of AM facilities in today’s remaining unexplored areas and enlarge the production series at lower costs.

## Figures and Tables

**Figure 1 polymers-14-01351-f001:**
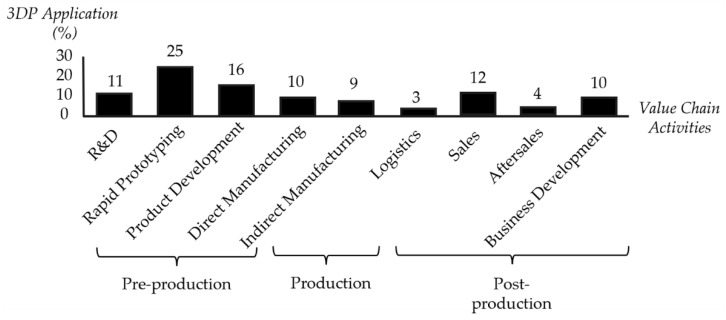
Percentages of 3DP applications according to the manufacturing function used in the surveyed companies. Reproduced with permission from reference [5].

**Figure 2 polymers-14-01351-f002:**
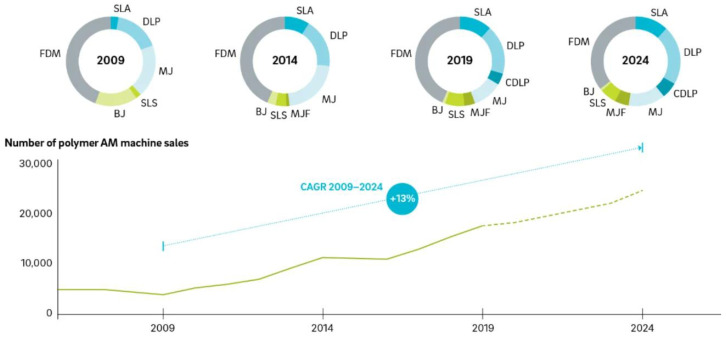
Evolution of the AM machine sales during the last 15 years. (Source: https://www.rolandberger.com/en/Point-of-View/Polymer-additive-manufacturing-Market-today-and-in-the-future.html (accessed on 16 February 2022).

**Figure 3 polymers-14-01351-f003:**
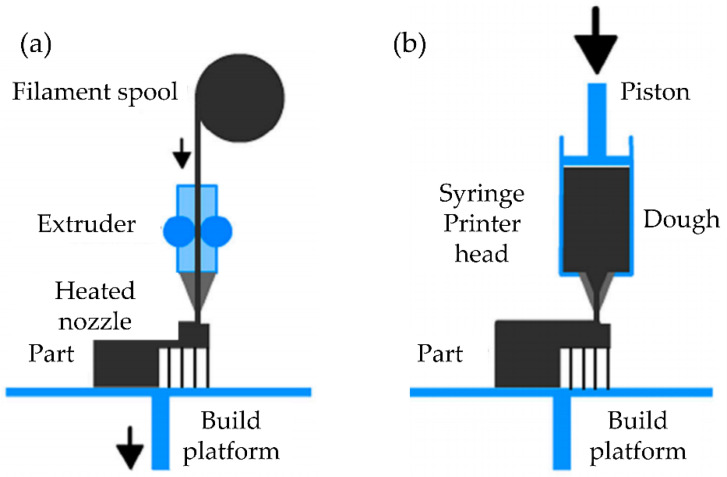
AM Alternatives involving material extrusion through a nozzle: (**a**) Fused deposition modeling. (**b**) Direct ink writing (e.g., bioprinting). Reproduced with permission from reference [20].

**Figure 4 polymers-14-01351-f004:**
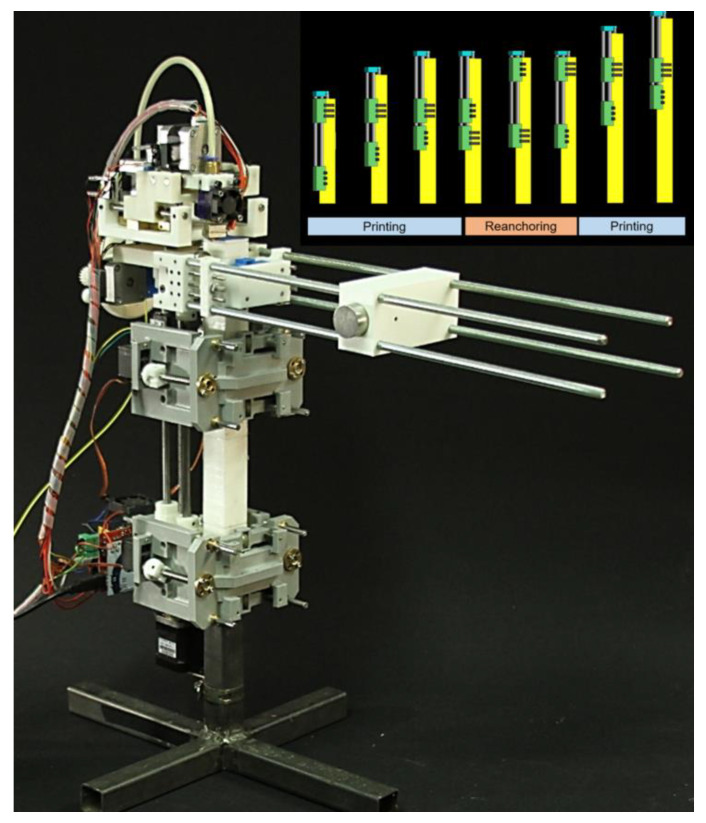
The Koala3D printing machine together with a schematic diagram of the printing and re-anchoring phases of the fabrication process. Reproduced with permission from reference [23].

**Figure 5 polymers-14-01351-f005:**
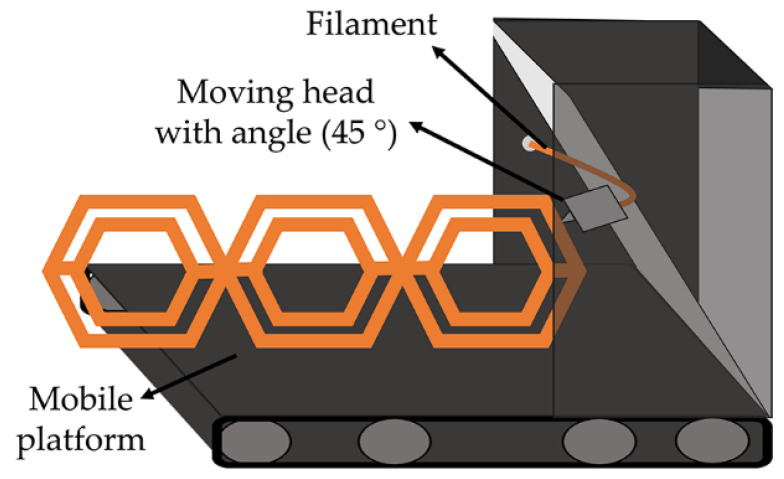
Schematic representation of continuous 3DP device.

**Figure 6 polymers-14-01351-f006:**
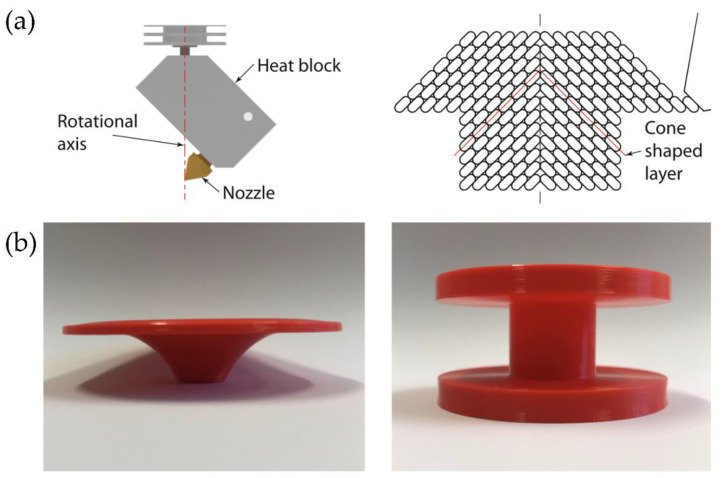
(**a**) Schematic representation of the 45° tilted nozzle and the conical layers. (**b**) Printed test parts using different shape-dependent strategies. Reproduced with permission from reference [33].

**Figure 7 polymers-14-01351-f007:**
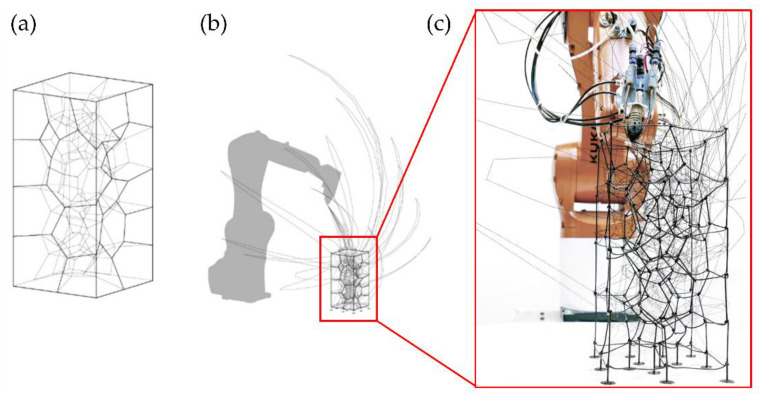
(**a**) 3D Voronoi design structure. (**b**) Robotic trajectories determined by Choreo. (**c**) Superposition of the final 3D-printed structure with the robotic arm trajectories. Reproduced with permission from reference [35].

**Figure 8 polymers-14-01351-f008:**
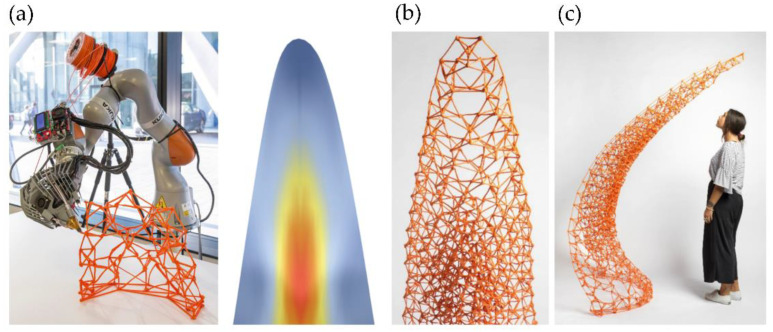
(**a**) The structure being printed by a robot. (**b**) The structural analysis of the 3D-printed part under self-weight stress. (**c**) The 2m-high self-standing prototype. Reproduced with permission from reference [34].

**Figure 9 polymers-14-01351-f009:**
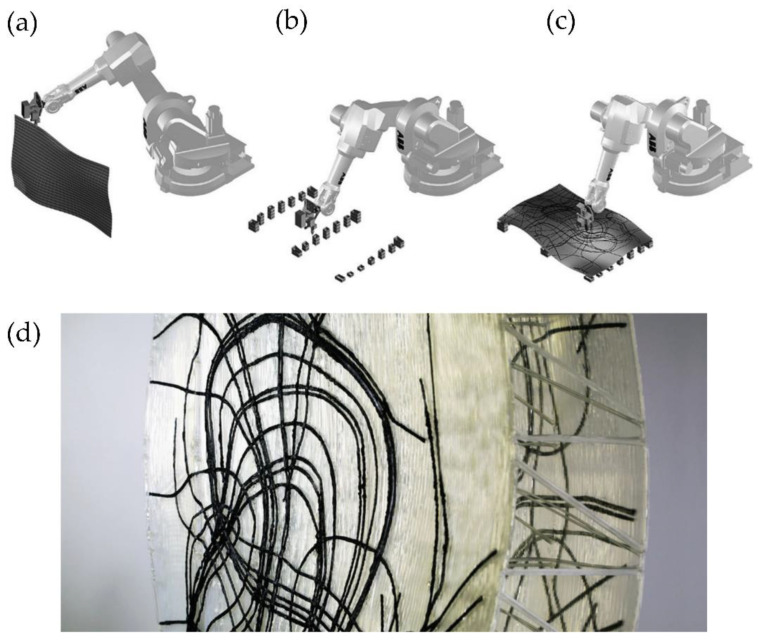
Steps followed to fabricate the façade CFRP prototype: (**a**) 3DP of the plastic base structure. (**b**) 3DP of plastic temporary supports and vertical repositioning. (**c**) Continuous CFRP add-on 3DP. (**d**) Photograph of the final 3D-printed part. Reproduced with permission from reference [36].

**Figure 10 polymers-14-01351-f010:**
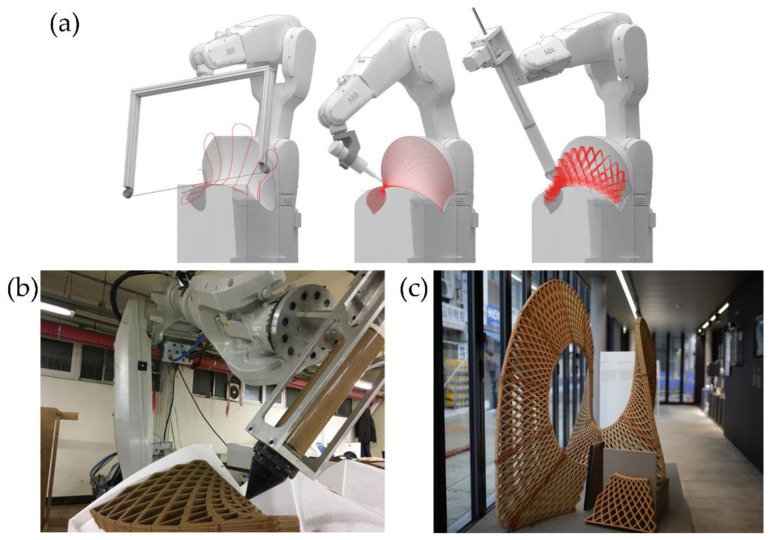
(**a**) Schematic representation of the robotic arm utilizing the three different methodologies (hotwire cutter, spindle, and clay extruder). (**b**) Clay 3D printing on molds using IRB 6700. (**c**) Photograph of the resulting structure. Reproduced with permission from reference [37].

**Figure 11 polymers-14-01351-f011:**
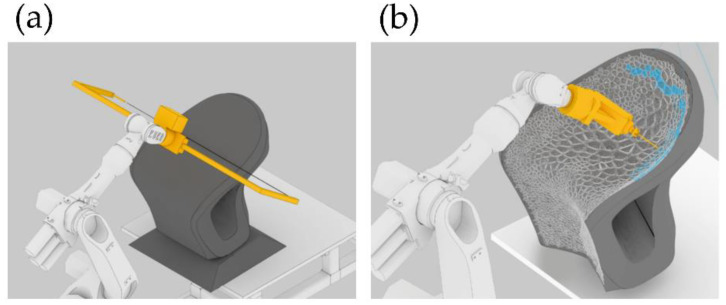
(**a**) Robotic hot wire cutting of overall form. (**b**) Robotic milling and silicone printing on the concave part of the prototype chair. Reproduced with permission from reference [38].

**Figure 12 polymers-14-01351-f012:**
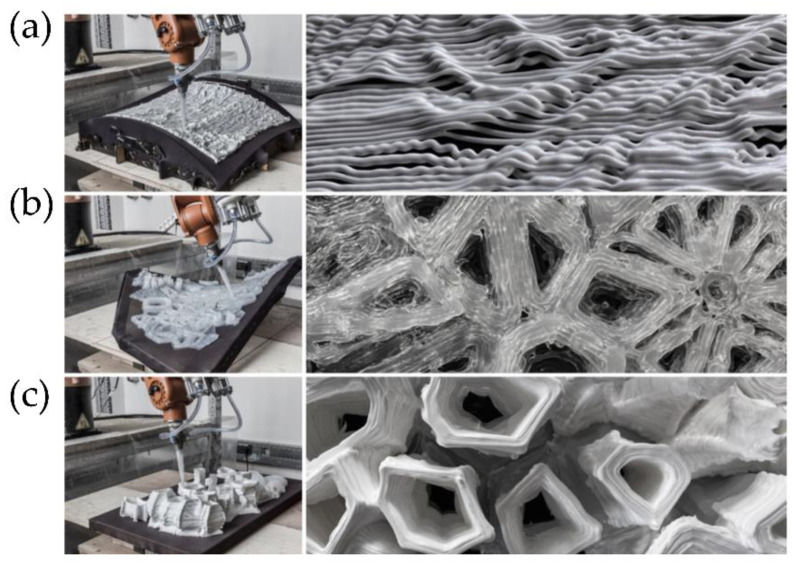
Robotic silicone printing experiments: (**a**) Linear continuous printing. (**b**) Cellular printing on a freeform fabric. (**c**) Prototype testing height, cantilevering, and size ranges. Reproduced with permission from reference [38].

**Figure 13 polymers-14-01351-f013:**
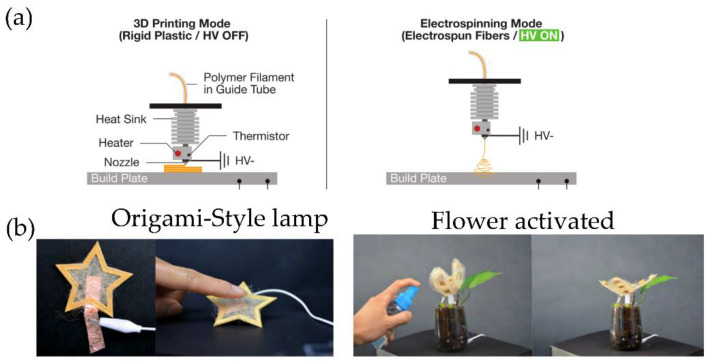
(**a**) A simplified representation of rigid plastic 3D printing and electrospinning. (**b**) A custom-shaped origami-style lamp printed using rigid plastic and electrospun textile a shaped flower made of electrospun textile and rigid plastic actuates based on the soil’s water level. Reproduced with permission from reference [39].

**Figure 14 polymers-14-01351-f014:**
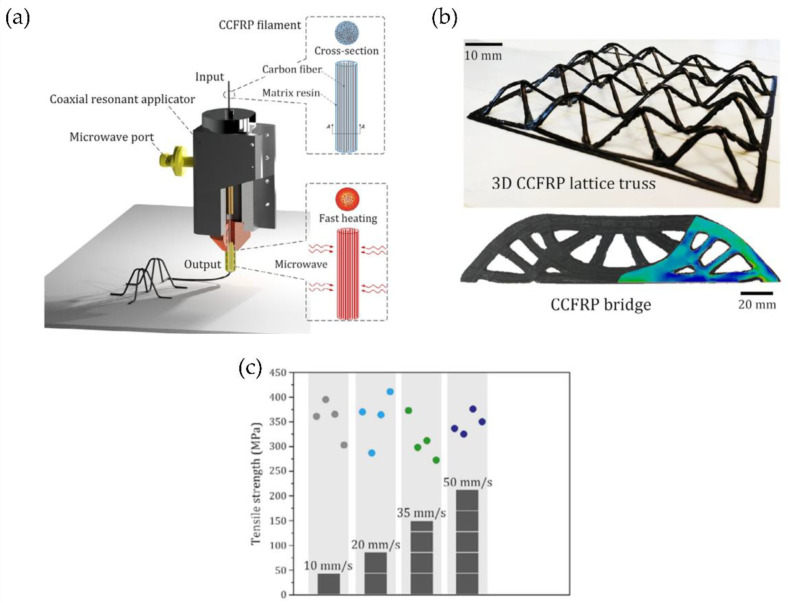
(**a**) Schematic diagram of the microwave printing head functioning. (**b**) CCFRP structures printed at high speed. (**c**) Tensile strength of the specimens printed at different speeds. Reproduced with permission from reference [40].

**Figure 15 polymers-14-01351-f015:**
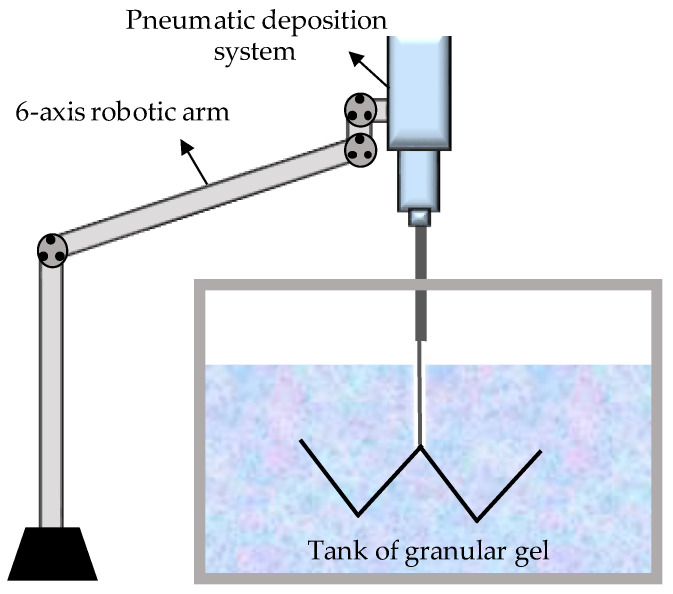
Schematic RLP system using a six-axis robotic arm and a pneumatic gun for the deposition.

**Figure 16 polymers-14-01351-f016:**
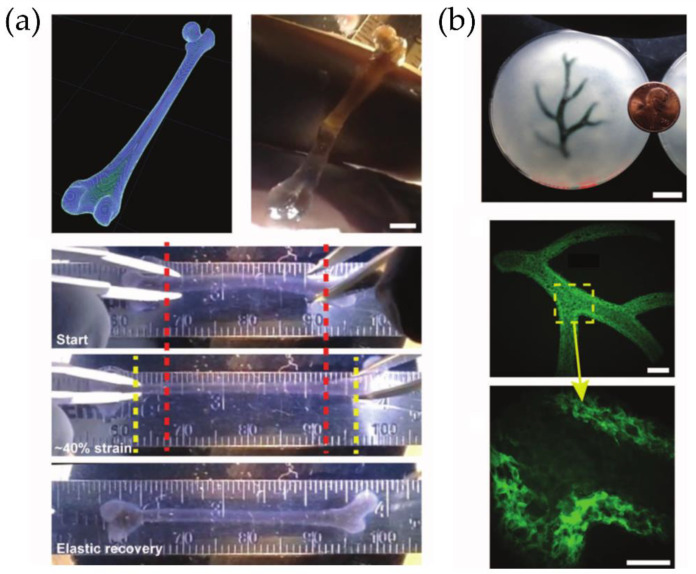
(**a**) A human femur model printed with FRESH in alginate after removal from the support bath. (**b**) An example of the arterial tree printed in alginate (black). A section of the arterial tree was printed from fluorescent alginate. Reproduced with permission from reference [48].

**Figure 17 polymers-14-01351-f017:**
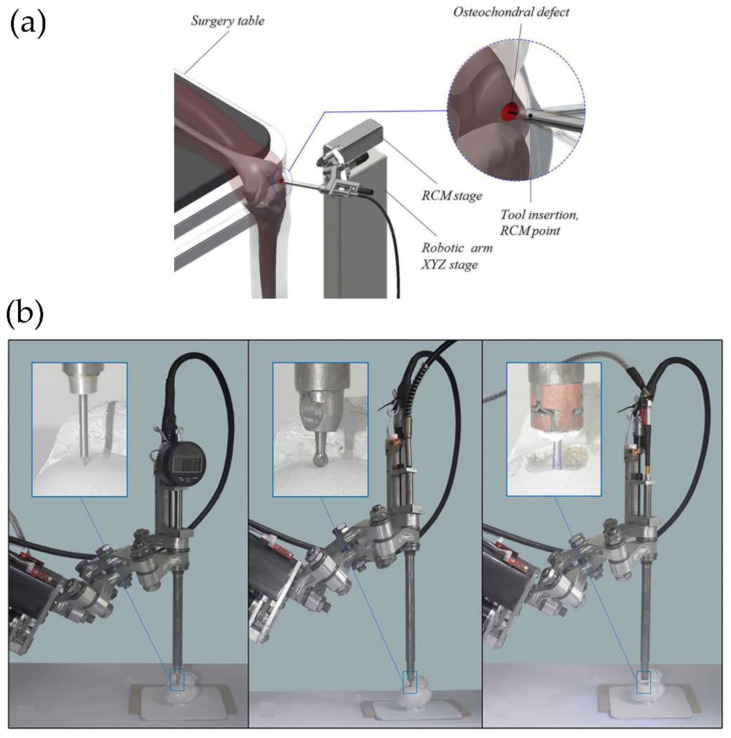
(**a**) Proposed surgery setup. (**b**) Experimental procedure for surface registration, bone milling, and 3D printing. Reproduced with permission from reference [55].

**Figure 18 polymers-14-01351-f018:**
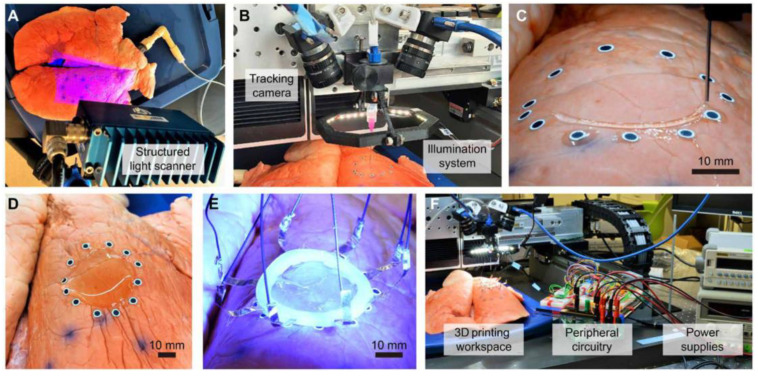
(**A**) 3D scanning of the porcine lung surface. (**B**) The custom-built 3D printing gantry system. (**C**) In situ 3D printing of hydrogel ink on a porcine lung. (**D**) The 3D-printed circular layer of a hydrogel. (**E**) UV light curing the hydrogel layer with the silicone ring and embedded electrodes. (**F**) The hardware setup for in situ monitoring of lung deformation with the printed EIT sensor. Reproduced with permission from reference [57].

**Figure 19 polymers-14-01351-f019:**
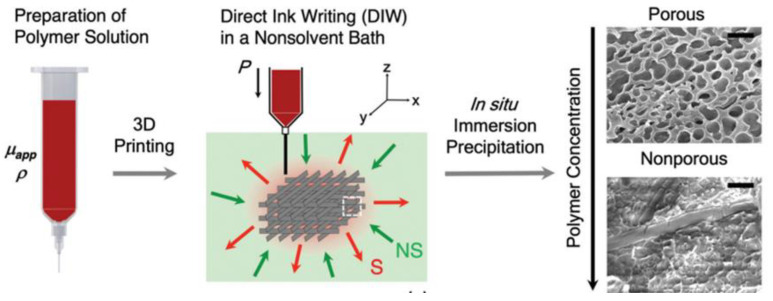
Overview of immersion precipitation 3D printing (Ip-3DP) and structures printed via this method. Reproduced with permission from reference [59].

**Figure 20 polymers-14-01351-f020:**
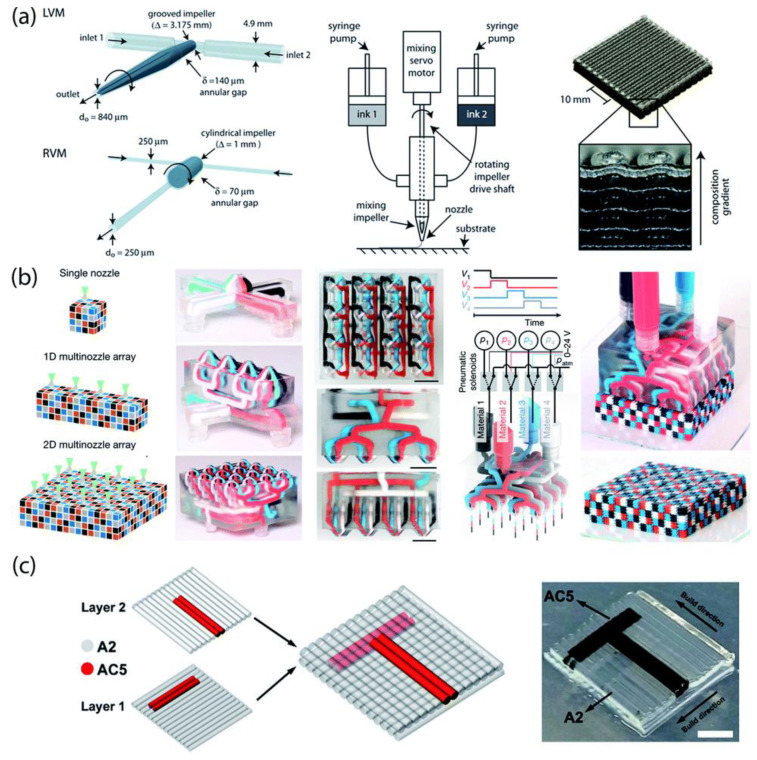
Multi-material 3D printing approaches for soft materials. (**a**) Systems for active mixing in situ during printing using rotating impellers (grey shadow). (**b**) Multi-material, multi-nozzle 3D printheads in a microfluidic system combining fast pneumatic solenoids and soft inks enable voxelated printing. (**c**) Deposition of hydrogel based on an aspiration-on-demand protocol. Reproduced with permission from reference [19].

**Figure 21 polymers-14-01351-f021:**
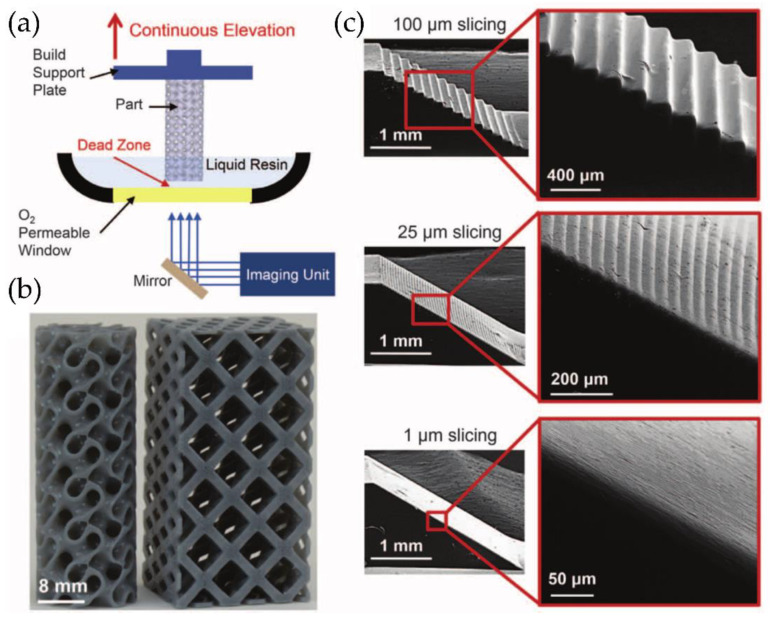
(**a**) Schematic representation of the CLIP technology. (**b**) Resulting printed parts obtained with this technology. (**c**) Ramp test patterns produced at the same printing speed with different slicing thicknesses (100, 25, and 1 μm). Reproduced with permission from reference [73].

**Figure 22 polymers-14-01351-f022:**
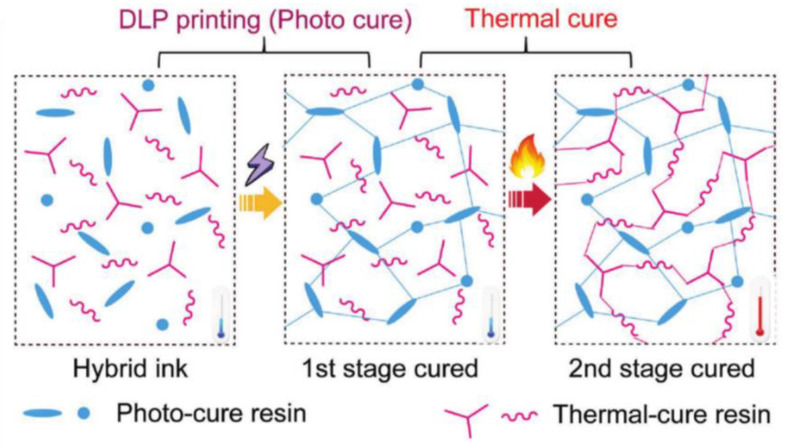
Schematic diagram explaining the two-stage curing process, photopolymerization via DLP followed by a thermopolymerization. Reproduced with permission from reference [76].

**Figure 23 polymers-14-01351-f023:**
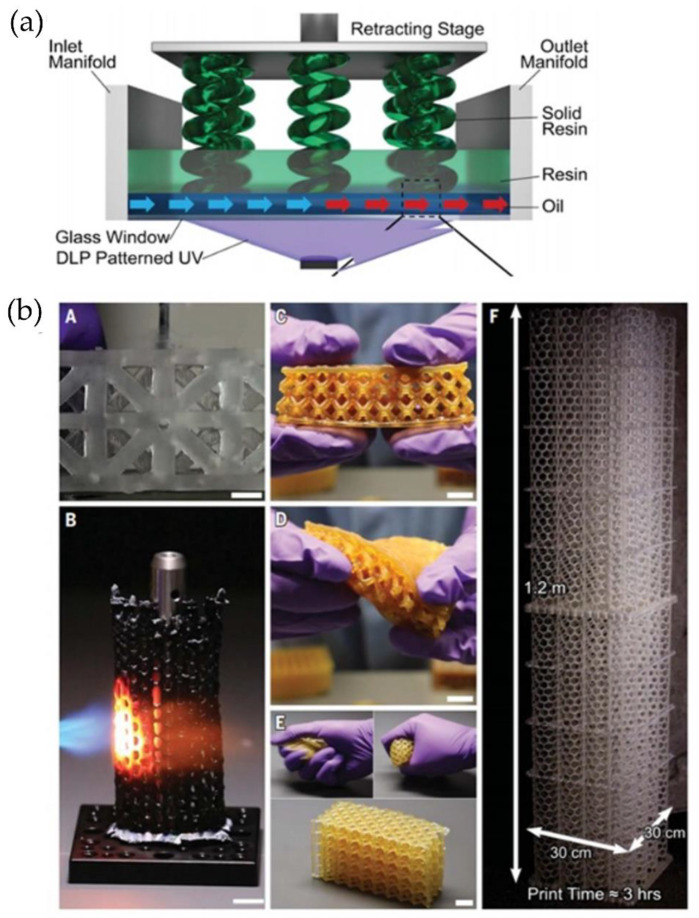
(**a**) Scheme of a 3D-printed part emerging from the resin vat using the HARP technology. (**b**) Photographs of a series of materials printed with HARP technology where A = A hard, machinable polyurethane acrylate part; B = A post-treated silicon carbide ceramic printed lattice; C,D = A printed butadiene rubber structure; E = Polybutadiene rubber returns to expanded lattice after compression and F = Hard polyurethane acrylate lattice printed. Reproduced with permission from reference [77].

**Figure 24 polymers-14-01351-f024:**
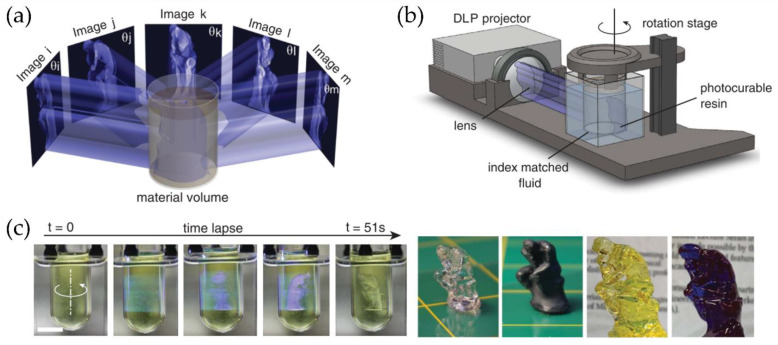
(**a**) Patterned illumination from many directions to create a computed 3D structure. (**b**) Schematic representation of the CAL system used. (**c**) Sequential view of the build volume during a CAL print. Several photographs of the resulting final part using different materials (as printed, printed in black for clarity, thermally cured version of the structure, and opaque version using violet crystals in the resin). Reproduced with permission from reference [82].

**Figure 25 polymers-14-01351-f025:**
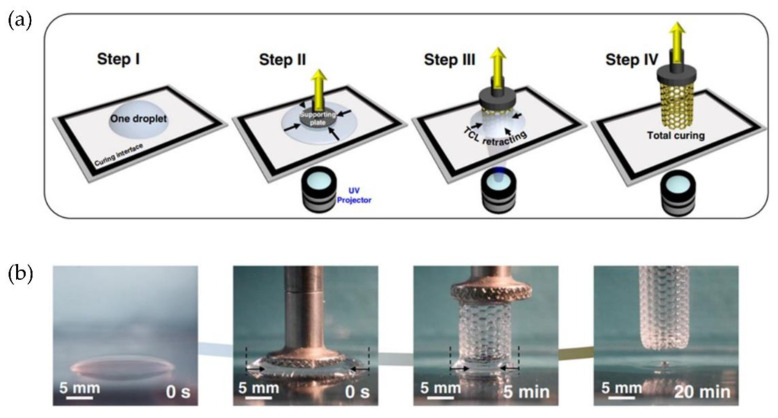
(**a**) Scheme of single-droplet resin curing into the desired 3D structures. (**b**) Sequence of optical images of the UV-curing process. Reproduced with permission from reference [83].

**Figure 26 polymers-14-01351-f026:**
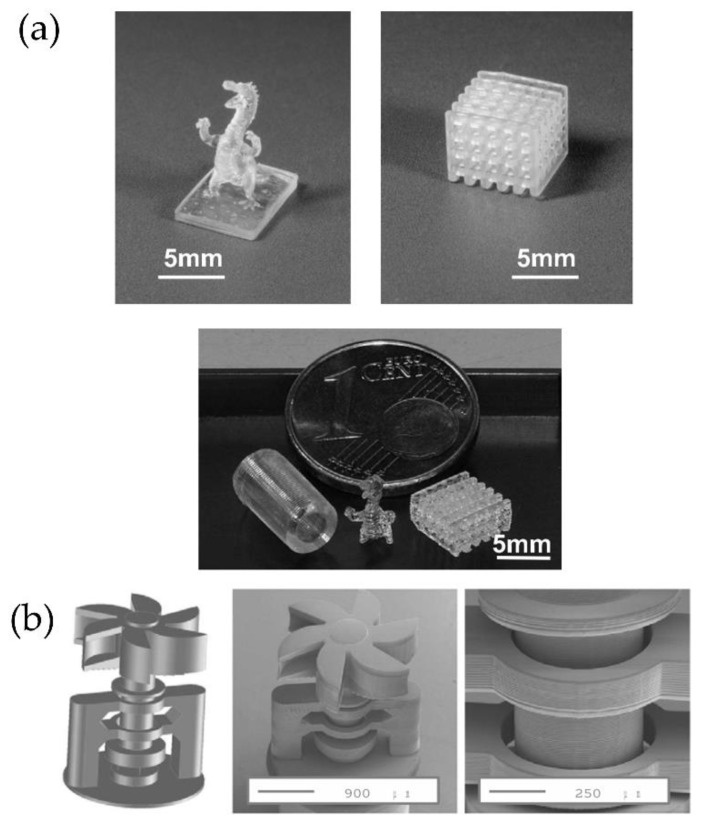
(**a**) Photographs of test parts made of PEGDA/UDMA with the PuSL system and CEA/PEGDA gels. (**b**) SEM image of a micromechanical arrangement with movable components. Reproduced with permission from reference [86].

**Figure 27 polymers-14-01351-f027:**
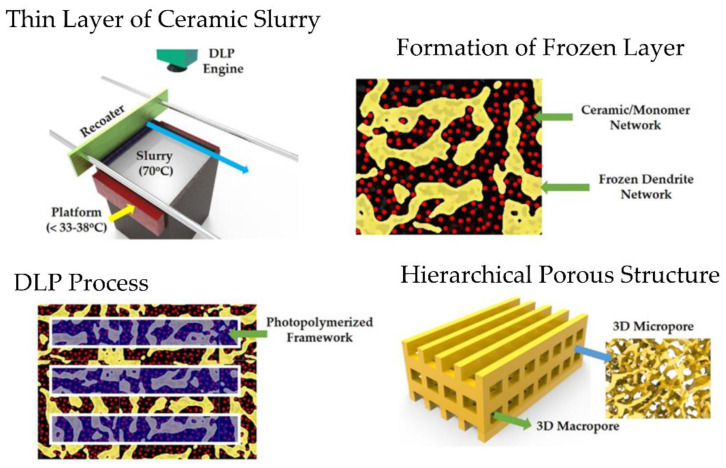
Schematic diagrams of the freeze-drying DLP process using a photopolymerizable ceramic slurry. This method produces a hierarchical macro/micro-porous 3D structure. Reproduced with permission from reference [89].

**Figure 28 polymers-14-01351-f028:**
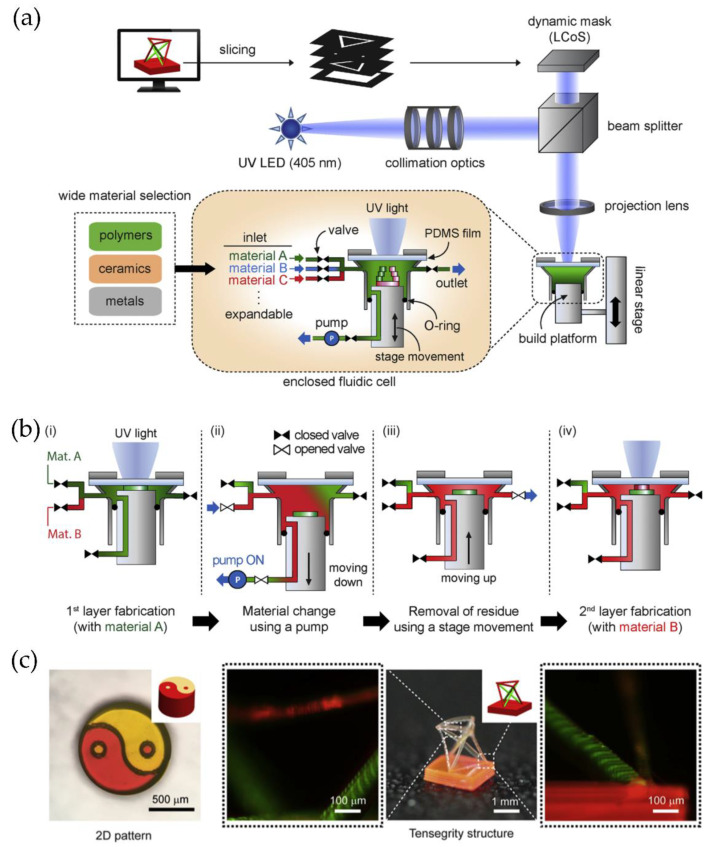
(**a**) Schematic illustration of the multi-material micro-SLA overall process. (**b**) Material exchange process. (**c**) Optical microscope image of the Taiji symbol patterned cylinder made of two different materials and fluorescent microscope images of a tensegrity structure consisting of multi-material high aspect ratio beams. Reproduced with permission from reference [92].

**Figure 29 polymers-14-01351-f029:**
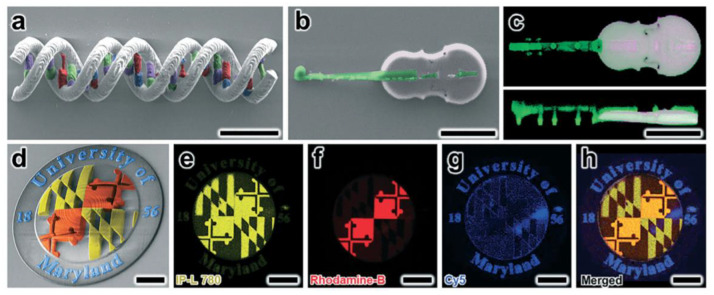
Results for various multi-material microstructures fabricated via DLW. (**a**) False-colored SEM results for a five-material DNA-inspired component. (**b**) SEM and (**c**) confocal fluorescence micrographs of two-material cello-inspired structures. (**d**–**h**) Overlapped SEM and confocal fluorescence micrographs of a four-material University of Maryland logo. Reproduced with permission from reference [93].

**Figure 30 polymers-14-01351-f030:**
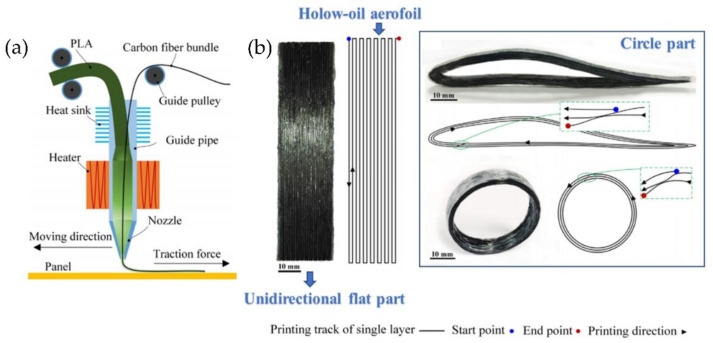
(**a**) Schematics of the designed extrusion device to printing continuous carbon fiber reinforced PLA. (**b**) 3D printing path of continuous carbon fiber reinforced PLA composite parts: unidirectional flat part, hollow-out aero foil, and circle part. Reproduced with permission from reference [105].

**Figure 31 polymers-14-01351-f031:**
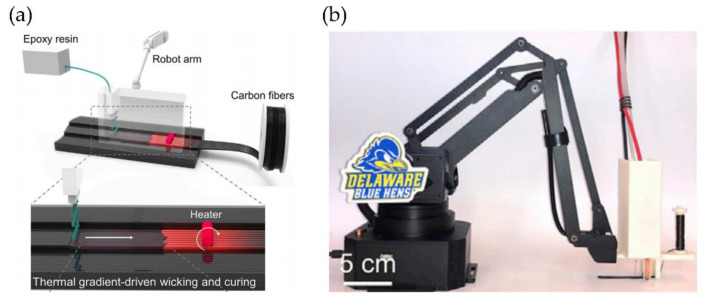
(**a**) Schematic representation of the LITA 3D printing approach. (**b**) LITA 3D printing system used in this work. Reproduced with permission from reference [107].

**Figure 32 polymers-14-01351-f032:**
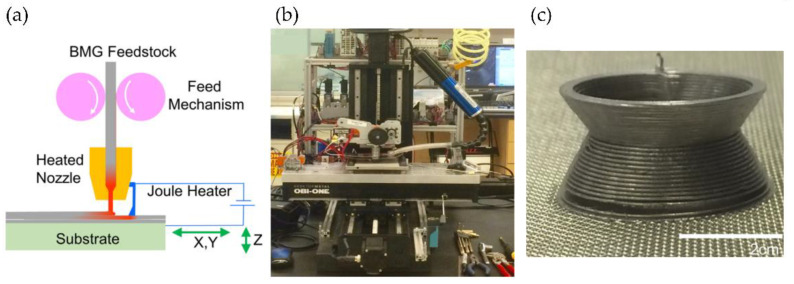
(**a**) Schematics of the FFF process for direct-write extrusion of BMGs. (**b**) Physical setup of the BMG printer. (**c**) BMG parts printed in continuous mode via the proposed method. Reproduced with permission from reference [125].

**Figure 33 polymers-14-01351-f033:**
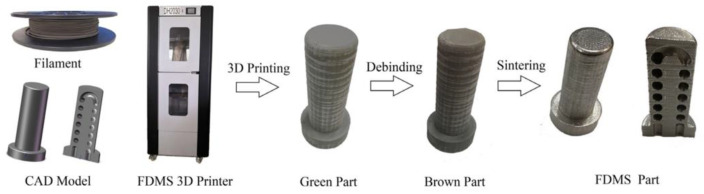
Schematic illustration of FDMS process. Reproduced with permission from reference [127].

**Figure 34 polymers-14-01351-f034:**
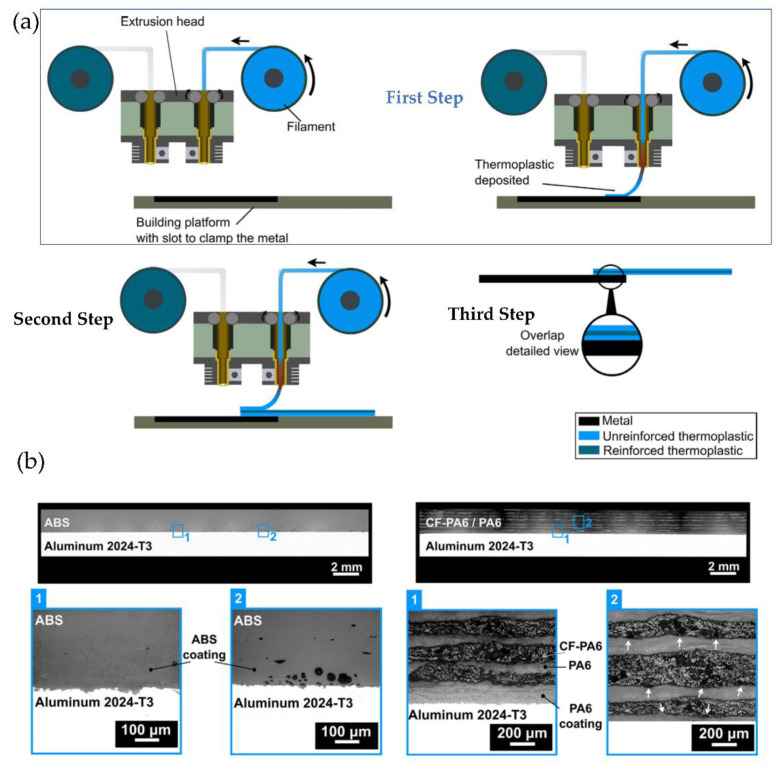
(**a**) Schematic representation of the AddJoining process. (**b**) Cross-sectional microstructure of hybrid joints for aluminum 2024-T3/ABS and aluminum 2024-T3/PA6/CF-PA6. Reproduced with permission from reference [129].

**Figure 35 polymers-14-01351-f035:**
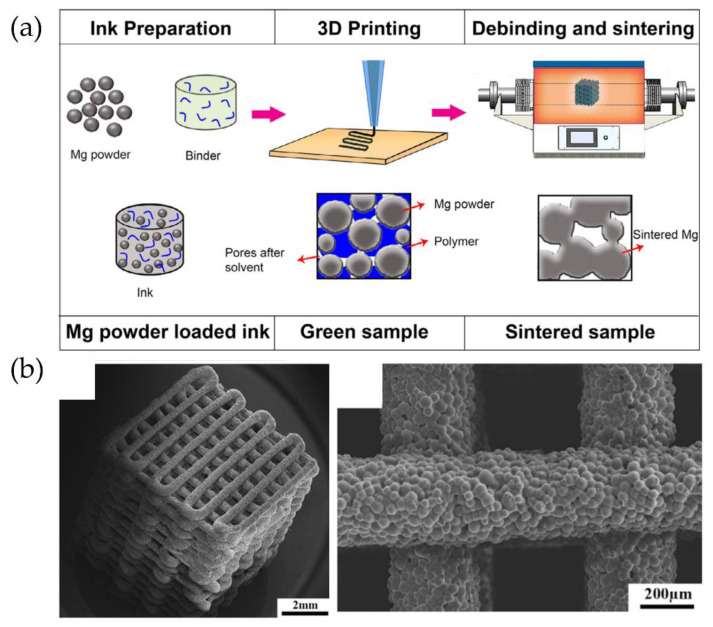
(**a**) A schematic diagram of the fabrication steps and the designed structure of Mg scaffolds. (**b**) SEM images of scaffolds of Mg printed by SC3DP. Reproduced with permission from reference [134].

**Figure 36 polymers-14-01351-f036:**
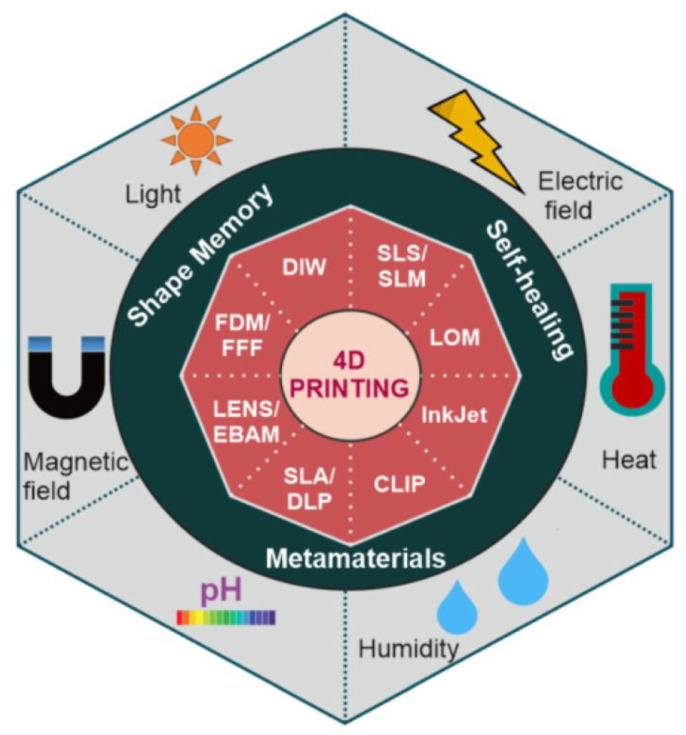
Schematic overview of 4D printing advances, including technologies, materials, and stimuli. Reproduced with permission from reference [135].

**Figure 37 polymers-14-01351-f037:**
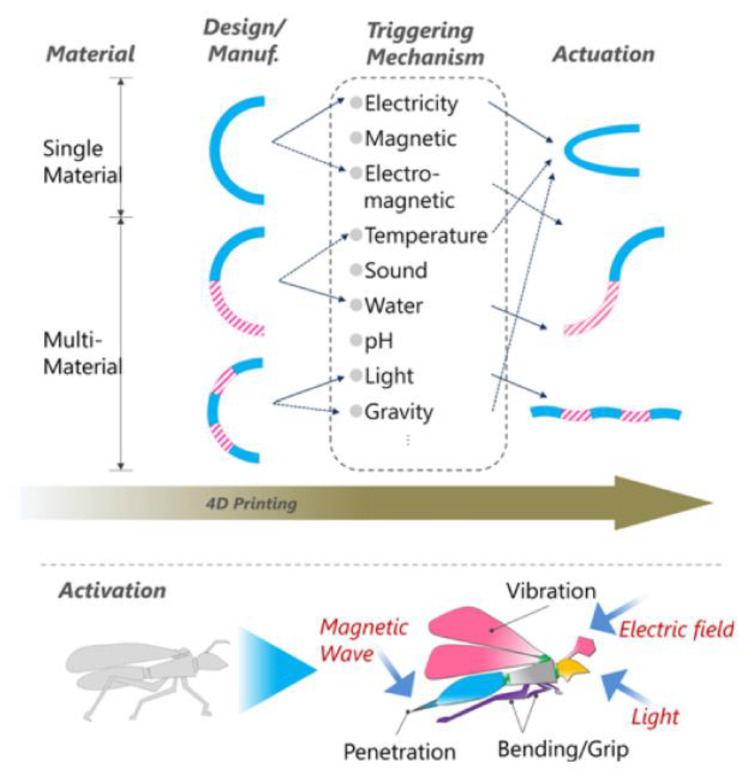
**Top**: Schematic illustration of the mechanism proposed to achieve actuation and shape flexibility using single or multi-stimuli responsive materials. **Bottom**: Design of an artificial bug controlled by multi-stimuli activation. Reproduced with permission from reference [136].

**Figure 38 polymers-14-01351-f038:**
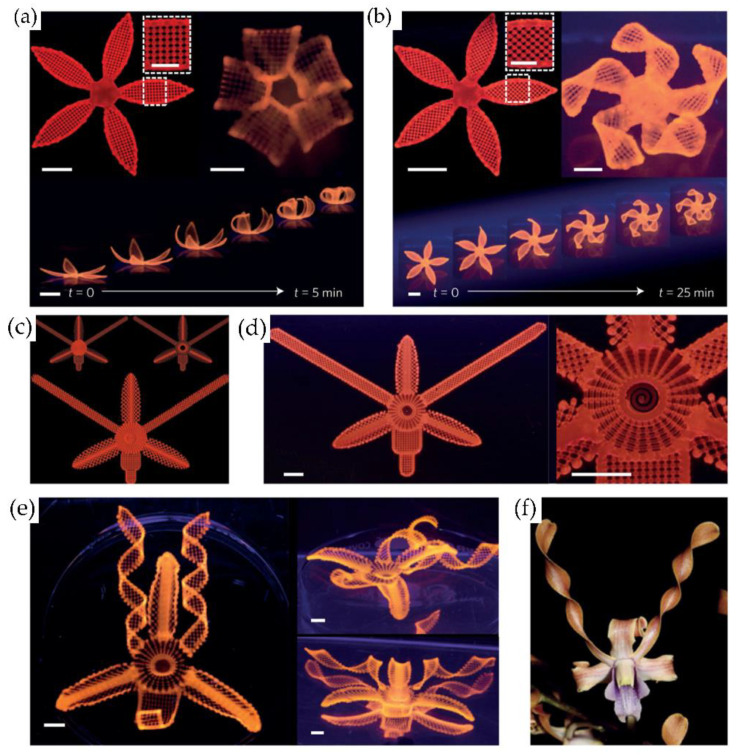
Complex flower morphologies generated by biomimetic 4D printing. (**a**,**b**) Simple flowers composed of 90°/0° and −45°/45° bilayers oriented concerning the long axis of each petal, (**c**–**f**) print path (**c**), printed structure (**d**) and resulting swollen structure (**e**) of a flower demonstrating a range of morphologies inspired by a native orchid, the Dendrobium helix (courtesy of Ricardo Valentin) (**f**). Reproduced with permission from reference [137].

**Figure 39 polymers-14-01351-f039:**
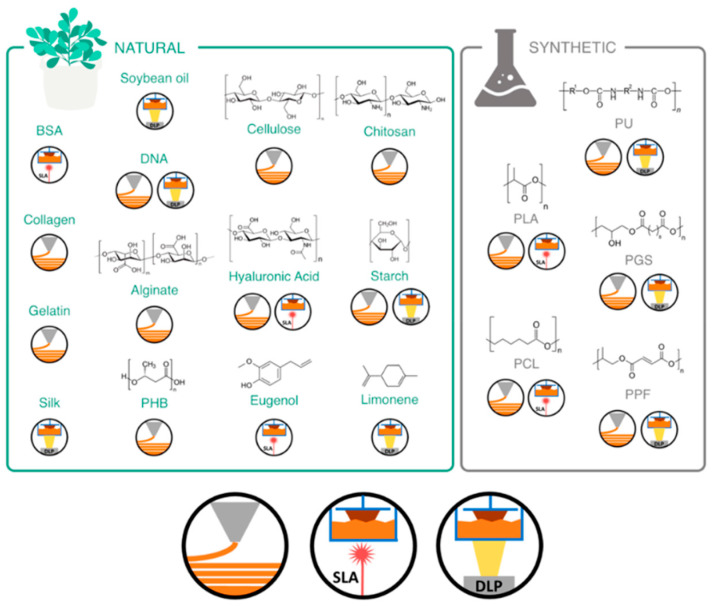
Some renewable feedstocks were developed for sustainable AM. Natural: bovine serum albumin (BSA), collagen, gelatin, silk, soybean oil, DNA, alginate, PHB, cellulose, hyaluronic acid, eugenol, chitosan, starch, and limonene. Synthetic: polyurethane (PU), poly(lactic acid) (PLA), poly(glycerol sebacate) (PGS), polycaprolactone (PCL), and poly(propylene fumarate) (PPF). Reproduced with permission from reference [138].

**Table 1 polymers-14-01351-t001:** Current significant limitations of the most extensively employed AM technologies.

	Material Extrusion (FFF)	VAT Photopolymerization (SLA/DLP)
Resolution	X-Y: Above 150 microns (generally 400 microns). Z: Above 50 mm (usually, 100–200 microns).	X-Y: Laser (SLA): 140–160 microns. UV light (DLP): 50–60 microns. Z: As low as 20 microns (usually 50–100 microns).
Continuous (multipart)/discontinuous	Typically, discontinuous.	Discontinuous. Even discontinuous in the fabrication layer by layer.
Size limit	Tens of cm up to meter scale.	Generally, between 20–50 cm (X, Y, and Z).
Part anisotropy	High.	Low.
Free 3D fabrication	Not allowed. Fabrication in a plane layer by layer.	Not allowed. Fabrication in a plane layer by layer.
Supports	Yes.	Yes.
Cost	Low.	Low-moderate.
Materials	Thermoplastics, elastomers, composites, and viscoelastic pastes.	Thermosets, elastomers, and composites.

**Table 2 polymers-14-01351-t002:** Summary of advantages and disadvantages of FFF and SLA processes. Reproduced with permission from reference [15].

Process	Advantages	Disadvantages
Material Extrusion (FFF)	Low cost of the entry-level machines. A variety of raw materials are available. Versatile and easy to customize.	Low level of precision and long build time. Unable to build sharp external corners. Anisotropic nature of a printed part.
Vat Photopolymerization (SLA/DLP)	High-resolution and accuracy, good surface finish. High fabrication speed. Low-imaging specific energy.	Require post-processing to remove support. Require post-curing for enhanced strength. Limited range of materials.

## Data Availability

Not applicable.

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
