# Peer review of "Innovation in Additive Manufacturing Using Polymers: A Survey on the Technological and Material Developments"

_polymers, 2022, doi:10.3390/polym14071351_

Round 1

Reviewer 1 Report

In this manuscript, Mauricio Sarabia-Vallejos and coworkers summarize the most recent inovations in additive manufacturing using polymers. A detailed overview of the efforts made to improve the two most extensively employed techniques, i.e., material extrusion and VAT-photopolymerization, are presented.

This manuscript will be of interest to the Readers of Polymers after major revision. There are several concerns the authors need to address. These are listed below:

  • The text does not compare the two techniques, does not list the pros and cons, advantages and disadvantages;
  • In the article, it would be desirable to specify the polymers that can be used for material extrusion or VAT-photopolymerization. Ideally this information should be provided in table;
  • There is no information on compatibility of the fillers with polymers and their impact on final product (pages 30-31)
  • We recommend adding some missing references about polymers systems and fillers that are perspective for additive manufacturing:

https://doi.org/10.1016/j.colsurfa.2022.128525

https://doi.org/10.3762/bjnano.10.233

https://doi.org/10.1016/j.addma.2021.101844

  • There are also minor corrections that need to be made to the manuscript. These include the following:
  1. On page 1, line 5 change “7”to “5”;
  2. On page 4, line 117 put “of” before “applications”;
  3. On page 7, line 199 change the font of the signature to figure 5;
  4. On page 14, line 390 change the font;
  5. On page 39, line 1104 put a full stop at the end of the sentence.

Therefore, I believe the target journal is an appropriate forum for this article after major revision.

Author Response

Reviewer 1:

In this manuscript, Mauricio Sarabia-Vallejos and coworkers summarize the most recent inovations in additive manufacturing using polymers. A detailed overview of the efforts made to improve the two most extensively employed techniques, i.e., material extrusion and VAT-photopolymerization, are presented.

This manuscript will be of interest to the Readers of Polymers after major revision. There are several concerns the authors need to address. These are listed below:

  • The text does not compare the two techniques, does not list the pros and cons, advantages and disadvantages.

Answer: Indeed, as the reviewer comments, in the manuscript, we performed a comparison between additive manufacturing technologies and standard manufacturing but never made a direct comparison between the advantages and disadvantages of FFF and SLA. This discussion has been included in the manuscript at the end of section 2 as text and as a new table. Thanks for your commentary.

  • In the article, it would be desirable to specify the polymers that can be used for material extrusion or VAT-photopolymerization. Ideally this information should be provided in table.

Answer: As the reviewer commented, this is an exciting topic, but as this is not the review's main focus, we just added a few lines about this at the end of section 2. Thanks for your commentary.

  • There is no information on compatibility of the fillers with polymers and their impact on final product (pages 30-31). We recommend adding some missing references about polymers systems and fillers that are perspective for additive manufacturing:

https://doi.org/10.1016/j.colsurfa.2022.128525

https://doi.org/10.3762/bjnano.10.233

https://doi.org/10.1016/j.addma.2021.101844

Answer: Thanks for your commentary, a short paragraph about the influence of fillers and their impact in printed part final properties was added in the text. The references you mentioned were added together with different ones to complement the discussion. Thank you very much for your comment as it helps us greatly to improve the quality of the article.

  • There are also minor corrections that need to be made to the manuscript. These include the following:

  1. On page 1, line 5 change "7" to "5";

  1. On page 4, line 117 put "of" before "applications";

  1. On page 7, line 199 change the font of the signature to figure 5;

  1. On page 14, line 390 change the font;

  1. On page 39, line 1104 put a full stop at the end of the sentence.

Answer: All of these changes were performed in the manuscript. Thanks for the commentary.

Reviewer 2 Report

The paper is well-written as a very comprehensive survey of polymers in additive manufacturing. To be more complete the authors should include powder bed fusion processes for polymers like Selective Laser Sintering (SLS), including the variety of materials in research and used in the industry. One example is the powder material filled with Cu, Zi, Ag and other bacteriostatic componentes for medical, and industrial applications. 

Author Response

Reviewer 2:

The paper is well-written as a very comprehensive survey of polymers in additive manufacturing. To be more complete, the authors should include powder bed fusion processes for polymers like Selective Laser Sintering (SLS), including the variety of materials in research and used in the industry. One example is the powder material filled with Cu, Zi, Ag, and other bacteriostatic components for medical, and industrial applications.

Answer: Thanks for your commentary. To be honest, in the first draft of this bibliographic review, we did include the topic of SLS as the third most important technology in AM, but because the article was too long, we preferred to remove it from the manuscript. But, due to your comment, we have decided to add a couple of paragraphs summarizing the study we had previously done on this topic (section 5). Thank you very much for your comment as it helps us greatly to improve the quality of the article.

Reviewer 3 Report

The present review paper deals with 3D printing. There are many similar reviews. It is well written and discussed, there are no significant weaknesses.

While minor comments could be connected with the up to date FFF/SLA printing of biobased polymers. This information is missing in the text. There are many papers dealing within. For example, you could reference and also many others are also avaiable - 10.1021/acsami.8b13031;
10.3390/polym13081195;10.3390/polym11010116;  10.1016/j.polymdegradstab.2020.109347; 10.3390/polym13081195

Author Response

Reviewer 3:

The present review paper deals with 3D printing. There are many similar reviews. It is well written and discussed, there are no significant weaknesses.

While minor comments could be connected with the up to date FFF/SLA printing of biobased polymers. This information is missing in the text. There are many papers dealing within. For example, you could reference and also many others are also avaiable:

 10.1021/acsami.8b13031

10.3390/polym13081195

10.3390/polym11010116

10.1016/j.polymdegradstab.2020.109347

Answer: Thanks for your commentary, contribution like these helps us greatly to improve the quality of the article. We add the references that you mention in the section 2 of the manuscript as complementary materials used for SLA printing derived from vegetable oils.